SciPost Physics

Submission

# Dissipation-Induced Steady States in Topological Superconductors: Mechanisms and Design Principles

M.S. Shustin[1,2*], S.V. Aksenov[1,2], I.S. Burmistrov[1,3]

**1** L.D. Landau Institute for Theoretical Physics, acad. Semenova av.1-a, Chernogolovka 142432, Russia
**2** Kirensky Institute of Physics, Federal Research Center KSC SB RAS, 660036 Krasnoyarsk, Russia
**3** Laboratory for Condensed Matter Physics, HSE University, Moscow, 101000, Russia
*mshustin@yandex.ru

August 25, 2025

## Abstract

The search for conditions supporting degenerate steady states in nonequilibrium topological superconductors is important for advancing dissipative quantum engineering, a field that has attracted significant research attention over the past decade. In this study, we address this problem by investigating topological superconductors hosting unpaired Majorana modes under the influence of environmental dissipative fields. Within the Gorini-Kossakowski-Sudarshan-Lindblad framework and the third quantization formalism, we establish a correspondence between equilibrium Majorana zero modes and non-equilibrium kinetic zero modes. We further derive a simple algebraic relation between the numbers of these excitations expressed in terms of hybridization between the single-particle wavefunctions and linear dissipative fields. Based on these findings, we propose a practical recipes how to stabilize degenerate steady states in topological superconductors through controlled dissipation engineering. To demonstrate their applicability, we implement our general framework in the BDI-class Kitaev chain with long-range hopping and pairing terms — a system known to host a robust edge-localized Majorana modes.

# Notations

- Lowercase characters will denote scalars: real and complex.

- Capital letters denote matrices, the dimension of which is clear from the context.

- Symbols with caps denote operators operating in the Fock space. For example, $\hat{H}$ is a many-body Hamiltonian of the system, and $\hat{\rho}$ is a many-body density matrix. Moreover, in the Fock-Liouville operator space $\mathcal{K}$, such operators (for example, $\hat{P}$) are vectors and are sometimes denoted as $|\hat{P}\rangle$.

- Standard bra and ket symbols denote vectors in the Fock space (for example, $|1\rangle$), while symbols containing operators (for example, $|\hat{P}\rangle$) stand for vectors in the Fock-Liouville space.

- Superoperators operating in the Fock-Liouville space are indicated by symbols with inverted caps. For example, $\check{c}_j$ and $\check{c}_j^+$ are fermionic superoperators satisfying the canonical anticommutation relations $\{\check{c}_i, \check{c}_j^+\} = \delta_{ij}$, $\{\check{c}_i, \check{c}_j\} = 0$ in the space $\mathcal{K}$. At the same time, depending on the context, such operators act either in Fock space, mapping operators to operators: $\check{A} : \hat{P} \to \hat{P}$, or in the Fock-Liouville space $\mathcal{K}$, mapping vectors to vectors: $\check{A} : |\hat{P}\rangle \to |\hat{P}\rangle$.

- Underlined symbols represent vectors of the corresponding scalar or operator values. For example, $\underline{\hat{w}}$ is a column vector of Majorana operators acting in the Fock space.

# 1 Introduction

Progress in the experimental implementation of cold atom systems and quantum optics has attracted considerable attention to dissipative engineering [1–8] of quantum low-dimensional systems [9–11]. Within that framework, complex many-body states are realized due to the controlled action of the external dissipative environment on the subsystem under consideration. At the same time, the realized non-equilibrium steady states (NESS) can differ significantly from equilibrium ones, demonstrate exotic effects and non-intuitive phase transitions, see e.g. Refs. [12–20]. If the dissipative environment affects a nontrivial topological phase, the realized NESS might also acquire features of topological order and exhibit protection to external perturbations [21–27]. If the density matrix corresponding to such a state is pure, then the so-called dark state [1, 2] is realized; in the degenerate case such steady states become the dark space. There is an interest in the dark states and spaces as promising objects for storing, deleting, and transmitting quantum information [1, 28–32].

The main theoretical tool of NESS research is the analysis of stationary solutions of the Gorini-Kossakowski-Sudarshan-Lindblad (GKSL) equation, the most general equation for the density matrix of the system under study, in which memory of interaction with the environment is neglected [33–35]. The effect of the environment is modeled by introducing the so-called Lindblad (or jump) operators which describe one-, two-, etc. processes of particle transfer from the subsystem to the environment and back. Effective theoretical methods have been developed to describe the steady states of the GKSL equation. They include the Keldysh contour functional integration method [9, 10, 36] and the third quantization formalism [37–40]. It has been shown that for free bosons or fermions (implying that jump operators are linear in terms of fermionic or bosonic creation/annihilation operators), the description of the dynamics governed by the GKSL equation is equivalent to solving a non-Hermitian single-particle problem.

An important issue of dissipative quantum engineering is the NESS dimension analysis. In the case of the GKSL equation, a series of theorems were proved defining algebraic conditions for the Hamiltonian of the system and jump operators for which the stationary solution is unique [41–46]. Further interest in the formation of dark spaces led to the emergence of theorems that set sufficient conditions for existence of degenerate NESS. So, for example, it was shown that if the GKSL equation has the so-called strong symmetries, given the existence of unitary operators with which both the Hamiltonian of the system and the jump operators commute, the number of NESS is determined by the number of invariant subspaces of operators of such symmetries [47–51]. An analog of the Lieb-Schultz-Mattis theorem has recently been proved for open systems with translational invariance and strong $U(1)$ symmetry, which states that it is impossible to implement a single NESS with an odd total charge of the system [52]. We emphasize that the described necessary conditions for the existence of degenerate NESS require a special class of dissipative fields: either reflecting the symmetry of the Hamiltonian, or corresponding to a homogeneous system preserving the total charge.

Recently, Majorana states, possessing spatial non-locality and obeying non-Abelian exchange statistics, have been actively studied for potential applications in topological quantum computing [53–58]. Considering this, as well as the existing problems with the detection of Majorana states [59, 60], it is natural to ask about the implementation of the phase of topological superconductivity within dissipative dynamics. In particular, the relevant question is to what extent the interaction of topological superconductors with the external environment

can contribute to the implementation of Majorana modes suitable for quantum computing, and when and how these interactions act in the opposite direction. We note that as in the non-dissipative case, such issues are mainly focused on the Kitaev chain model [53] and its generalizations.

Certain results have been achieved in the study of the dark states in topological superconductors with linear dissipative fields, in which the jump operators of the GKSL equation are linear combinations of the fermion creation and annihilation operators. In Ref. [61–63], the problem of obtaining an explicit form of linear Lindblad operators for a Kitaev chain tunnel coupled with several equilibrium Fermi reservoirs was considered. Also jump operators were derived for description of the low-energy dynamics of the so-called Majorana box qubit [64,65], i.e. two Majorana nanowires connected by a superconducting bridge, whose Majorana modes interact with quantum dots [66]. The possibility of realizing dark states and spaces in such a system that have common properties with the spaces of qubits formed by Majorana excitations has been demonstrated. In this case, the characteristics of such states can be controlled by the gate voltages at the quantum dots. It was later shown that in systems containing several Majorana box qubits, nontrivial braiding of non-equilibrium Majorana modes can be achieved [67] .

It was shown in [68–71] that non-equilibrium Majorana modes can also occur in an ensemble of localized fermions, those hybridization is provided by dissipative fields alone (there is no Hamiltonian in the GKSL description of the evolution of the system density matrix). To achieve the regime of a symmetric point of the Kitaev chain, a special type of Lindblad operators was used, the structure of which is given by gapped Bogolyubov excitations at a special point of the model. Also, for such a case (a dissipative system without unitary dynamics), a topological invariant based on the properties of a Gaussian density matrix was proposed, the values of which allows predicting the existence of non-equilibrium Majorana modes [70]. In addition, it was noted in Ref. [70] that each Lindblad operator, not being Hermitian or anti-Hermitian, in a general case leads to hybridization of a pair of Majorana modes, reducing the multiplicity of NESS degeneracy by two (a fact that is also shown in the present work). However, no analysis of how one might affect the dimensionality of NESS has been performed.

It is worth noting that dissipative topological superconductors are sometimes described in the framework of the non-Hermitian Hamiltonian approach, which considers only the Hamiltonian and dissipative dynamics of the density matrix by neglecting the quantum jump term of the GKSL equation. As well-known [72–79], the latter induces stochastic processes in the system. Within the framework of non-Hermitian Hamiltonian approach, the spectrum of excitations of the Kitaev chain was studied and the conditions for the occurrence of stable zero modes that share properties similar to Majorana modes were analyzed. It has been shown that non-unitary evolution of the density matrix expands significantly the conditions for the realization of such excitations. However, it should be noted that taking into account the quantum jump term in the GKSL equation may be of fundamental importance for describing some of the observed processes or the effect of generalized measurements. For example, stochastic quantum trajectories manifest themselves in the noise statistics of measured quantities [80], can lead to phase transitions in the entanglement [14–16], as well as to quantum diffusion in the case of intense environmental impact on the subsystem [19, 81–84, 86].

In this paper, we study the spectral properties of topological superconductors (TS) affected by dissipation and/or in the presence of coupling to the equilibrium reservoirs. The evolution of the density matrix of the system is modeled in the framework of the GKSL equation, in

which the unitary evolution of TS is described by Hamiltonian which is quadratic in fermionic operators, while jump operators are linear in fermionic operators (linear dissipative fields). To analyse the system we use the third quantization method with the introduction of the Fock-Liouville operator space. The main goal of our study is to analyze the evolution of Majorana zero modes (MM) of topological superconductors when exposed to linear dissipative fields. We study the most general formulation of the problem: an arbitrary TS Hamiltonian hosting several Majorana modes and arbitrary linear dissipative fields. We emphasize that in contrast to Refs. [47–49, 80], we do not impose internal symmetries on the system considered to guarantee degenerate NESS. The latter is the consequence of non-trivial topological phases in the isolated system.

We demonstrate that the MM in the isolated TS are transformed into the so-called zero kinetic modes (ZKM) in the presence of dissipation as well as coupling to reservoirs. These ZKM have strictly zero energies and have properties in common with MM: their excitation leads to a change in the structure of the density matrix, which is indistinguishable from the point of view of local operators. It is why such ZKM are considered by us as analogues of MM in open systems. Importantly, the number of ZKM ($N_0$) is smaller than the number of MM ($2N_M$) in the isolated TS. We prove that $N_0 = 2N_M - \mathrm{rk}\,\mathcal{B}$ where the hybridization matrix $\mathcal{B} \in \mathbb{R}^{2N_M \times 2N_B}$, cf. Eq. (53), describes the hybridization of $2N_M$ MM wave functions of the isolated TS with $N_B$ dissipative fields describing dissipative baths and/or reservoirs. It is worthwhile to mention that depending on the rank of the matrix $\mathcal{B}$ the number of ZKM may be odd. We demonstrate that existence of the ZKM has one to one correspondence with the strong symmetry of the Lindbladian realized by the unitary operator. Based on the knowledge of the hybridization matrix $\mathcal{B}$, we propose practical recipes that allow manipulating the number of NESS in dissipative TS.

We apply our general results to the generalized Kitaev chain with long-range hoppings and superconducting pairings. In such model MM localized at the edges of the chain can be easily constructed. We illustrate how the proposed recipes allow to control the number and the structure of the ZKM. Also, we show that dissipation can be used to manipulate the ZKM, e.g. transferring them from one edge of the chain to the other.

The outline of the paper is as follows. We formulate our dissipative model in Sec. 2. In Sec. 3 we review the third quantization method in application to the considered model. The conditions for existence of ZKM as well as corresponding strong symmetry are derived in Sec. 4. Our general results applied to Kitaev chain are described in Sec. 5. In Sec. 6 we end the paper with conclusions. Some technical details are delegated to the Appendix 7.

## 2 Formulation of the model

### 2.1 The model of dissipative dynamics

In this section, we formulate a model of superconductors that are affected by dissipative fields that do not preserve the number of fermions. As we demonstrate in Appendix 7, coupling to reservoirs can be also described as the action of the dissipative fields. Therefore, in what follows, we do not consider coupling to the reservoirs separately. We assume that the evolution of the density matrix of the system affected by $N_B$ dissipative channels satisfies the following

GKSL equation:

$$\frac{d\hat{\rho}}{dt} = \hat{\mathcal{L}}\,\hat{\rho} = -i\,[\hat{H},\,\hat{\rho}] + \sum_{v=1}^{N_B} \left( 2\hat{L}_v \hat{\rho}\hat{L}_v^+ - \hat{L}_v^+ \hat{L}_v \hat{\rho} - \hat{\rho}\hat{L}_v^+ \hat{L}_v \right), \tag{1}$$

where the unitary evolution is governed by a generic quadratic Hamiltonian for spinless fermions:

$$\hat{H} = i \sum_{j,k=1}^{2N} A_{jk}\,\hat{w}_j\,\hat{w}_k = i\,\underline{\hat{w}}^T A\,\underline{\hat{w}}, \qquad A_{jk} = A_{jk}^* = -A_{kj}. \tag{2}$$

Here and further, following Refs. [37–39], the underscore $\underline{x} = (x_1,\,x_2,\,\ldots)^T$ means a vector of the corresponding quantities. The Hamiltonian (2) is written in terms of Majorana operators satisfying $\hat{w}_j = \hat{w}_j^+$, $\{\hat{w}_j,\,\hat{w}_k\} = 2\delta_{jk}$ $(j, k = 1,\ldots, 2N)$. These Majorana operators can be written in terms of fermionic creation $(\hat{c}_l^+)$ and annihilation operators $(\hat{c}_l)$:

$$\hat{w}_{2l-1} = \hat{c}_l + \hat{c}_l^+, \qquad \hat{w}_{2l} = i(\hat{c}_l - \hat{c}_l^+), \qquad l = 1,\ldots, N. \tag{3}$$

It is convenient to consider a system on a lattice and associate the index $l = 1,\ldots, N$ with lattice sites. The Hamiltonian (2) corresponds to a system of non-interacting fermions or a system of weakly interacting fermions in the mean field approximation [10]. The operators $\hat{L}_v$ describe the system's coupling to the $v$-th dissipative channel and are assumed to be linear in Majorana operators:

$$\hat{L}_v = \sum_{j=1}^{2N} l_{v,j}\,\hat{w}_j = \underline{l}_v \cdot \underline{\hat{w}} = \sum_{l=1}^{N} \left[ \mu_{v,l}\,\hat{c}_l + \nu_{v,l}\,\hat{c}_l^+ \right],$$

$$\mu_{v,l} = l_{v,\,2l-1} - i\,l_{v,\,2l}, \qquad \nu_{v,l} = l_{v,\,2l-1} + i\,l_{v,\,2l}. \tag{4}$$

Hereafter, $\underline{a} \cdot \underline{b}$ denotes the dot product of two vectors. Here $l_{v,j}$ are arbitrary complex numbers. Such type of jump operator describes the transfer of fermions from and to the system. In this approximation, each dissipative channel is described by two real dissipative fields:

$$\underline{l}_v^{r} = (\operatorname{Re}l_{v,1}, \operatorname{Re}l_{v,2}, \ldots, \operatorname{Re}l_{v,2N})^T, \qquad \underline{l}_v^{i} = (\operatorname{Im}l_{v,1}, \operatorname{Im}l_{v,2}, \ldots, \operatorname{Im}l_{v,2N})^T. \tag{5}$$

However, if the amplitudes of gain and loss are complex conjugated to each other, $\nu_{v,l} = \mu_{v,l}^*$, i.e. $\hat{L}_v^+ = \hat{L}_v$, then the $v$-th dissipative channel is actually described by a single dissipative field $\underline{l}_v^{r}$, since $\underline{l}_v^{i} \equiv 0$.

## 2.2   Solution for the unitary evolution

The use of the Majorana energy representation assumes the representation of the real skew-symmetric matrix $A$ in the canonical form:

$$A_c = \mathcal{W}^T A\,\mathcal{W} = \bigoplus_{a=1}^{N} \begin{pmatrix} 0 & \varepsilon_a \\ -\varepsilon_a & 0 \end{pmatrix}; \quad \mathcal{W}\mathcal{W}^T = \mathcal{W}^T\mathcal{W} = I_{2N}, \tag{6}$$

where the orthogonal matrix

$$\mathcal{W} = [\underline{\chi}_1,\; \underline{\chi}_2, \cdots,\; \underline{\chi}_{2a-1},\; \underline{\chi}_{2a}, \cdots,\; \underline{\chi}_{2N-1},\; \underline{\chi}_{2N}] \in \mathbb{R}^{2N \times 2N}. \tag{7}$$

is composed from the $2N$ orthonormal vectors, $\underline{\chi}_a \cdot \underline{\chi}_b = \delta_{ab}$. The eigenvalues

$$\varepsilon_a = \underline{\chi}_{2a}^T A \underline{\chi}_{2a-1}, \qquad a = 1, \ldots, N \tag{8}$$

have the meaning of energies of Bogoliubov quasiparticles in the isolated TS. The operators of the creation and annihilation of Bogoliubov excitations can be represented in the standard form:

$$\hat{\alpha}_a^+ = \sum_{l=1}^{N} \left( v_{al}^* \hat{c}_l + u_{al}^* \hat{c}_l^+ \right), \qquad \hat{\alpha}_a = \sum_{n=1}^{N} \left( u_{al} \hat{c}_l + v_{al} \hat{c}_l^+ \right). \tag{9}$$

In order to relate $\underline{u}_a$ and $\underline{v}_a$ with $\underline{\chi}_a$, it is convenient to compose a pair of conjugate Majorana operators in the energy representation from the operators of the creation and annihilation of Bogoliubov excitations:

$$\hat{b}_a' = \hat{\alpha}_a + \hat{\alpha}_a^+ = \underline{\chi}_{2a-1} \cdot \underline{\hat{w}}, \qquad \hat{b}_a'' = i(\hat{\alpha}_a^+ - \hat{\alpha}_a) = \underline{\chi}_{2a} \cdot \underline{\hat{w}}, \qquad a = 1, \ldots, N, \tag{10}$$

where $(l = 1, \ldots, N)$

$$\begin{aligned}
\chi_{2a-1,\, 2l-1} &= \mathrm{Re}(u_{al} + v_{al}), & \chi_{2a-1,\, 2l} &= \mathrm{Im}(u_{al} - v_{al}), \\
\chi_{2a,\, 2l-1} &= \mathrm{Im}(u_{al} + v_{al}), & \chi_{2a,\, 2l} &= \mathrm{Re}(v_{al} - u_{al}).
\end{aligned} \tag{11}$$

Therefore, the real vectors $\underline{\chi}_{2a-1}$ and $\underline{\chi}_{2a}$ can be interpreted as wave functions of Bogoliubov excitations in the Majorana representation. We note that the Hamiltonian (2) can be written in terms of quasiparticle operators as

$$\hat{H} = 4 \sum_{a=1}^{N} \varepsilon_a (\alpha_a^+ \alpha_a - 1/2) \tag{12}$$

Below we will focus on the case in which there are $N_M$ zeroes among the set $\{\varepsilon_a\}$, i.e. $\varepsilon_a = 0$ for $a = 1, \ldots, N_M$. It is worthwhile to mention that we do not distinguish between zeroes that are not protected against small perturbation of the matrix $A$ and zeroes which existence is protected by the topology. The latter situation occurs in the case of TS. Such topologically protected MM correspond typically to the wave functions $\underline{\chi}_{2a-1}$ and $\underline{\chi}_{2a}$ that are localized in different spatial regions. Consequently, for any local (on the lattice) single-particle operator in the Majorana representation characterized by the skew-symmetric matrix $O$ the following relations hold $\underline{\chi}_{2a}^T O \underline{\chi}_{2a-1} \simeq 0$ at least in the limit $N \to \infty$. In the following sections, we will study a fate of the zero-energy excitations of $\hat{H}$ in the presence of dissipation.

# 3 Third quantization approach to solution of the GKSL equation

## 3.1 The Fock-Liouville space

In this section we discuss solution of the GKSL equation (1) similar to the description of the unitary evolution in Sec. 2.2. A convenient tool for this task is the formalism of the third

quantization developed in Refs. [37–39,87]. A key element of this formalism is the introduction of the so-called *Fock-Liouville space*, $\mathcal{K}$, defined as the Hilbert space with a linear hull

$$| \hat{P}_{\underline{\alpha}} \rangle = \hat{P}_{\alpha_1, \alpha_2, \ldots, \alpha_{2N}} = 2^{-N} \hat{w}_1^{\alpha_1} \hat{w}_2^{\alpha_2} \ldots \hat{w}_{2N}^{\alpha_{2N}}, \qquad \alpha_j \in \{0, 1\}, \tag{13}$$

and the scalar product in the Hilbert-Schmidt form:

$$\langle \hat{P}_{\underline{\beta}} | \hat{P}_{\underline{\alpha}} \rangle = \mathrm{Tr} \left( \hat{P}_{\underline{\beta}}^+ \cdot \hat{P}_{\underline{\alpha}} \right), \tag{14}$$

where the trace acts on the Fock space with dimensionality $2^N$.

In the definition (13), the parameters $\alpha_1, \alpha_2, \ldots, \alpha_{2N}$ have the meaning of occupation numbers in the $\mathcal{K}$ space. Therefore the vectors $| \hat{P}_{\alpha_1, \alpha_2, \ldots, \alpha_{2N}} \rangle$ is sometimes conveniently denoted as $| \alpha_1, \ldots, \alpha_{2N} \rangle$. It is worth noting that the number of occupation numbers in the Fock-Liouville space is $2N$ that is twice the number in the Fock space ($N$). This fact is related with the following: each fermionic degree of freedom corresponds to a pair of Majorana operators involved in the vectors (13). The vectors (13) form an orthonormal basis with respect to a scalar product (14), $\langle \hat{P}_{\underline{\beta}} | \hat{P}_{\underline{\alpha}} \rangle = \delta_{\underline{\alpha}\underline{\beta}}$. Next, we can define left and right Majorana superoperators, whose action on the elements of the $\mathcal{K}$ space leads to their multiplication by Majorana operators on the left and right, respectively [39]:

$$\check{w}_j^L | \hat{P}_{\underline{\alpha}} \rangle = | \hat{w}_j \cdot \hat{P}_{\underline{\alpha}} \rangle, \qquad \check{w}_j^R | \hat{P}_{\underline{\alpha}} \rangle = | \hat{P}_{\underline{\alpha}} \cdot \hat{w}_j \rangle,$$
$$\left[ \check{w}_i^L, \check{w}_j^R \right] = 0, \quad \{ \check{w}_i^L, \check{w}_j^L \} = \{ \check{w}_i^R, \check{w}_j^R \} = 2\delta_{ij}. \tag{15}$$

The algebra of such superoperators allows us to introduce fermionic superoperators $\check{c}_j$ and $\check{c}_j^+$ with the canonical anti-commutation relations $\{ \check{c}_i, \check{c}_j^+ \} = \delta_{ij}$, $\{ \check{c}_i, \check{c}_j \} = 0$:

$$\check{c}_j \hat{P}_{\underline{\alpha}} = -\frac{i}{2} \hat{W} \left[ \hat{w}_j, \hat{P}_{\underline{\alpha}} \right] = -\frac{i}{2} \hat{W} \left( \check{w}_j^L - \check{w}_j^R \right) \hat{P}_{\underline{\alpha}} = \frac{i}{2} \left( \check{w}_j^L + \check{w}_j^R \right) \hat{W} \hat{P}_{\underline{\alpha}},$$
$$\check{c}_j^+ \hat{P}_{\underline{\alpha}} = -\frac{i}{2} \hat{W} \{ \hat{w}_j, \hat{P}_{\underline{\alpha}} \} = -\frac{i}{2} \hat{W} \left( \check{w}_j^L + \check{w}_j^R \right) \hat{P}_{\underline{\alpha}} = \frac{i}{2} \left( \check{w}_j^L - \check{w}_j^R \right) \hat{W} \hat{P}_{\underline{\alpha}}. \tag{16}$$

Here we introduce the fermionic parity operator $\hat{W} = \hat{W}^+ = \exp(i\pi\hat{\mathcal{N}}) = i^N \prod_{j=1}^{2N} \hat{w}_j$ where $\hat{\mathcal{N}} = \sum_{l=1}^N \hat{c}_l^+ \hat{c}_l$ stands for the fermion number operator. It anti-commutes with any Majorana operator, $\{ \hat{W}, \hat{w}_j \} = 0$, and commutes with the Hamiltonian, $\left[ \hat{W}, \hat{H} \right] = 0$.

Next, inverting the relations (16), we represent the left and right Majorana superoperators in terms of fermionic ones:

$$\check{w}_j^L = -i \left( \check{c}_j^+ + \check{c}_j \right) \check{W} = i\check{W} \left( \check{c}_j^+ + \check{c}_j \right), \quad \check{w}_j^R = i \left( \check{c}_j^+ - \check{c}_j \right) \check{W} = i\check{W} \left( \check{c}_j^+ - \check{c}_j \right). \tag{17}$$

We note that the superoperator $\check{w}_j^{L,R}$ are non-local due to its dependence on $\check{W}$. However, since the physical observables are represented as products of an even number of operators $\hat{w}_j$, the superoperators of the physical observables in the space $\mathcal{K}$ are local and independent of $\check{W}$. [1] Then the mapping of operators from the Fock space to the Fock-Liouville space and vice versa is carried out using the following sequence of equalities:

$$\hat{w}_j \hat{P}_{\underline{\alpha}} = | \hat{w}_j \hat{P}_{\underline{\alpha}} \rangle = \check{w}_j^L | \hat{P}_{\underline{\alpha}} \rangle = i\check{W} \left( \check{c}_j^+ + \check{c}_j \right) | \hat{P}_{\underline{\alpha}} \rangle. \tag{18}$$

Similar sequence holds for the right superoperators $\check{w}_j^R$.

---

[1]We can always represent the product of an even number of superoperators $\check{w}_j^{L,R}$ from Eq. (17) in such a way that the non-local operator $\check{W}$ appears only as $\check{W}^2 = 1$. Therefore, the operators of local physical observables in the Fock-Liouville space can be constructed as local.

## 3.2   Mapping of GKSL equation to the Fock-Liouville space

In order to map the GKSL equation (1) to the Fock-Liouville space, the density matrix $\hat{\rho}$ should be considered as a vector, $\hat{\rho} \to |\hat{\rho}\rangle \in \mathcal{K}$, and the components of the Lindbladian $\hat{\mathcal{L}}$, the Hamiltonian $\hat{H}$ and jump operators $\hat{L}_v$, as the superoperators in the $\mathcal{K}$ space:

$$
\begin{aligned}
\hat{H}(\hat{\underline{w}})\hat{\rho} &\to \check{H}(\underline{\check{w}}_L)|\hat{\rho}\rangle, & \hat{L}_v^+(\hat{\underline{w}})\hat{L}_v(\hat{\underline{w}})\hat{\rho} &\to \check{L}_v^+(\underline{\check{w}}^L)\check{L}_v(\underline{\check{w}}^L)|\hat{\rho}\rangle, \\
\hat{\rho}\hat{H}(\hat{\underline{w}}) &\to -\check{H}(\underline{\check{w}}_R)|\hat{\rho}\rangle, & \hat{\rho}\hat{L}_v^+(\hat{\underline{w}})\hat{L}_v(\hat{\underline{w}}) &\to \check{L}_v(\underline{\check{w}}^R)\check{L}_v^+(\underline{\check{w}}^R)|\hat{\rho}\rangle, \\
\hat{L}_v(\hat{\underline{w}})\hat{\rho}\hat{L}_v^+(\hat{\underline{w}}) &\to \check{L}_v(\underline{\check{w}}^L)\check{L}_v^+(\underline{\check{w}}^R)|\hat{\rho}\rangle,
\end{aligned}
\tag{19}
$$

where the functional dependence of $\hat{H}(\hat{\underline{w}})$ and $\hat{L}(\hat{\underline{w}})$ is given by Eqs. (2) and (4), respectively.

Since the Majorana superoperators $\check{w}^{L,R}$ are expressed in terms of fermionic operators, $\check{c}$ and $\check{c}^+$, it is clear that (19) mappings can also be written in terms of them. As a result, in the third quantization approach, the stationary GKSL equation is reduced to an analog of the many-body Schrödinger equation:

$$
\hat{\mathcal{L}}\hat{\rho} = 0 \qquad \to \qquad \check{\mathcal{L}}(\check{c}, \check{c}^+)|\hat{\rho}\rangle = 0.
\tag{20}
$$

Here superoperator $\check{\mathcal{L}}(\check{c}, \check{c}^+)$ can be written in the Bogoliubov–de Gennes form:

$$
\check{\mathcal{L}} = -2\left(\underline{\check{c}}^{+T} \quad \underline{\check{c}}^T\right)\mathcal{L}\begin{pmatrix}\underline{\check{c}} \\ \underline{\check{c}}^+\end{pmatrix}, \qquad \mathcal{L} = \begin{pmatrix} X & Y \\ 0 & -X^T \end{pmatrix},
\tag{21}
$$

where the following notations are introduced

$$
X = -A + M_{\boldsymbol{r}}, \quad Y = -2iM_{\boldsymbol{i}}, \quad M = \sum_v \underline{l}_v \cdot \underline{l}_v^+, \quad M = M_{\boldsymbol{r}} + iM_{\boldsymbol{i}},
$$
$$
M_{\boldsymbol{r}} = \sum_v [\underline{l}_v^r \cdot \underline{l}_v^{rT} + \underline{l}_v^i \cdot \underline{l}_v^{iT}], \quad M_{\boldsymbol{r}} = M_{\boldsymbol{r}}^T, \quad M_{\boldsymbol{i}} = \sum_v [\underline{l}_v^i \cdot \underline{l}_v^{rT} - \underline{l}_v^r \cdot \underline{l}_v^{iT}], \quad M_{\boldsymbol{i}} = -M_{\boldsymbol{i}}^T.
\tag{22}
$$

The real part of the matrix $M$, $M_{\boldsymbol{r}}$ determines the dissipative channels and affects the spectrum of the Lindbladian. Its imaginary part, $M_{\boldsymbol{i}}$ contributes to the non-equilibrium correlation functions and determines how the system evolves to NESS along dissipative trajectories. The matrix $M_{\boldsymbol{r}}$ is positively semi-definite with non-negative eigen values.

It is worth noticing that the matrix $\mathcal{L}$ corresponding to the Liouvillian $\check{\mathcal{L}}$ in Eq. (21) has the size twice larger than the matrices involved in the GKSL equation. This is due to the fact that each fermionic degree of freedom corresponds to a pair of Majorana operators in the third quantization approach. Such doubling is similar to the one in the Keldysh field theory approach [10]. We note that the matrix $Y$ in Eq. (21) is contributed from all terms of dissipative part of the Liouvillian $\hat{\mathcal{L}}$. It can be constructed with the Keldysh field theory approach in which the corresponding matrix comes from the jump terms, $\hat{L}_v\rho\hat{L}_v^+$, alone. Such a difference occurs due to the usage of Majorana fermions in the third quantization approach and complex fermions in the Keldysh field theory approach. Both representations are related by the unitary rotation

$$
\mathcal{U} = \frac{1}{\sqrt{2}}\begin{pmatrix} I_N & I_N \\ -iI_N & iI_N \end{pmatrix}.
\tag{23}
$$

The triangular form of the matrix $\mathcal{L}$ suggests that the spectrum of the Liouvillian is determined by the eigenvalues of the matrix $X$. Below we assume that it can be diagonalized

as $U^{-1} X U = \mathrm{diag}(\underline{\beta}) \equiv B$ [^2]. Since $X + X^T = 2M_r$ the eigenvalues $\beta_l$ have non-negative real part, $\mathrm{Re}\,\beta_l \geq 0$. It guarantees that the evolution of the density matrix within the GKSL equation tends to a stationary state.

Since the matrix $X$ is real valued one, for each eigen value $\beta_l$ with $\mathrm{Im}\,\beta_l > 0$ there is the complex conjugated eigen value $\beta_{\bar{l}}$ with $\mathrm{Im}\,\beta_{\bar{l}} = -\mathrm{Im}\,\beta_l < 0$. The corresponding eigen vectors are related as:

$$\underline{U}_l \equiv \underline{\psi}_{2l-1} + i\underline{\psi}_{2l}, \qquad \underline{U}_{\bar{l}} = \underline{U}_l^* \equiv \underline{\psi}_{2l-1} - i\underline{\psi}_{2l} \tag{24}$$

and satisfy the relation $\underline{U}_{\bar{l}}^T A\,\underline{U}_l = -i\,\mathrm{Im}\,\beta_l\left(\underline{U}_{\bar{l}} \cdot \underline{U}_l\right)$ that is similar to Eq. (8). Eigen vectors $\underline{U}_l$ determines spatial behavior of the kinetic modes, $\mathrm{Im}\,\beta_l$ characterizes the frequency of the mode (similar to energies in the closed system), and $\mathrm{Re}\,\beta_l$ gives a rate of attenuation due to the presence of dissipation. In the case of isolated system, $\underline{\psi}_{2l-1} \to \underline{\chi}_{2l-1}$, $\underline{\psi}_{2l} \to \underline{\chi}_{2l}$, and a pair $(\beta_l, \beta_{\bar{l}}) \to \pm i\varepsilon_l$. In other words, in the absence of dissipation the spectrum of $X$ is situated on the imaginary axis of the plane $(\mathrm{Re}\,\beta, \mathrm{Im}\,\beta)$. For zero eigen values of $X$, $\beta_l = 0$ the eigen vectors $\underline{U}_l$ can be chosen to be real.

## 3.3   Spectrum of the Liouvillian

Let us introduce imaginary skew-symmetric *correlation matrix* $F = -F^* = -F^T$, which satisfies *Lyapunov-Sylvester equation* [88]:

$$X F + F X^T - Y = 0, \quad Y = -Y^* = -Y^T. \tag{25}$$

The above equation has the only solution for the matrix $F$ provided $\mathrm{Re}\,\beta_l > 0$ for all $l$. Physical meaning of $F$ is the non-equilibrium correlation function in Majorana representation: $F_{ij} = \frac{1}{2}\langle[\hat{w}_i,\hat{w}_j]\rangle$. Special care is needed if there exists the zero eigen values of $X$, $\beta_l = 0$, or pairs of purely imaginary eigen values, $\beta_l + \beta_{\bar{l}} = 0$. In this case the kernel of the operator $\mathcal{R}(F) = X F + F X^T$ is non-trivial. As it follows from the Fredholm alternative [89] a solution for $F$ exists if and only if the matrix $Y$ is orthogonal to the kernel of the operator $\bar{\mathcal{R}}(Z) = X^T Z + Z X$, i.e. $\mathrm{Tr}\left(Y^T Z\right) = 0$ for all solutions of the equation $\bar{\mathcal{R}}(Z) = 0$, see Theorem 1.1 of Ref. [90]. Let us consider a pair of eigenvalues with zero real part, $\beta_l$ and $\mathrm{Im}\,\beta_{\bar{l}} = -\mathrm{Im}\,\beta_l$. Then we can construct the solution for the equation $\bar{\mathcal{R}}(Z) = 0$ as $Z_{km} = U_{kl}(U_{lm}^{-1})^*$, where there is no summation over the index $l$. Then we find the following condition for the existence of solutions for $F$: $[U^{-1*}YU]_{ll} = 0$. Also, the obvious condition should hold $[U^{-1*}YU^{-1\,T}]_{ll} = 0$. In practice one can use the regularization $X \to X_\epsilon = X + \epsilon UU^{-1}$ with $\epsilon \to 0^+$ in order to find the solution of Eq. (25). We emphasize that such solution does not contain the information about ZKM and eigen states of $X$ with $\beta_{\bar{l}} = -\beta_l$. Also we note that this solution is not unique, since one can add to it the solution $\delta F$ of the homogeneous equation $X\delta F + \delta F X^T = 0$. We note that one can use any pair of eigen values with $\beta_{\bar{l}} = -\beta_l$ to construct the solution of the homogeneous equation as $\delta F^{(l\bar{l})} \sim \underline{U}_l\underline{U}_{\bar{l}}^T - \underline{U}_{\bar{l}}\underline{U}_l^T$.

With the help of the matrix $F$, the matrix $\mathcal{L}$ can be written in the diagonal form:

$$V\mathcal{L}V^{-1} = \begin{pmatrix} B & 0 \\ 0 & -B \end{pmatrix}, \quad V = \begin{pmatrix} U^{-1} & U^{-1}F \\ 0 & U^T \end{pmatrix}, \quad V^{-1} = \begin{pmatrix} U & -F(U^{-1})^T \\ 0 & (U^{-1})^T \end{pmatrix}. \tag{26}$$

---

[^2]: Since $A = -A^T$ and $M_r = M_r^T$ the matrix $X = A + M_r$ may be non-diagonalizable, but only reduced to a Jordan form. Below we consider such eigen vectors of the matrix $X$ that corresponds to eigen values with zero real part, $\mathrm{Re}\,\beta_l = 0$. For such eigen values, Jordan blocks are trivial due to the property $X + X^T \succeq 0$, see Lemma 2.3 of Ref. [38].

After such transformation the Liouvillian in Eq. (21) becomes

$$\check{\mathcal{L}} = -4\sum_{l=1}^{2N} \beta_l \check{b}_l^\times \check{b}_l\,; \quad \begin{pmatrix} \underline{\check{b}} \\ \underline{\check{b}}^\times \end{pmatrix} = V \begin{pmatrix} \underline{\check{c}} \\ \underline{\check{c}}^+ \end{pmatrix} = \begin{pmatrix} U^{-1}\,\underline{\check{c}} + U^{-1}\,F\,\underline{\check{c}}^+ \\ U^T\,\underline{\check{c}}^+ \end{pmatrix}. \tag{27}$$

Here the third quantized operators $\check{b}$ and $\check{b}^\times$ satisfy fermionic commutation relations: $\{\check{b}_l,\check{b}_m\} = \{\check{b}_l^\times,\check{b}_m^\times\} = 0,\ \ \{\check{b}_l,\check{b}_m^\times\} = \delta_{lm};\ \ l,m = 1,\ldots,2N$. We note that $\check{b}_l^\times \neq \check{b}_l^+$.

The diagonal representation (27) of the Liouvillian $\check{\mathcal{L}}$ allows us to determine its spectrum and to construct left and right eigenvectors in the $\mathcal{K}$ space:[3]

$$\check{\mathcal{L}}\,|\hat{\Theta}_{\underline{\eta}}^R\rangle = \lambda_{\underline{\eta}}\,|\hat{\Theta}_{\underline{\eta}}^R\rangle, \qquad \langle\hat{\Theta}_{\underline{\eta}}^L|\,\check{\mathcal{L}} = \lambda_{\underline{\eta}}\langle\hat{\Theta}_{\underline{\eta}}^L|, \qquad \lambda_{\underline{\eta}} = -4\sum_{j=1}^{2N}\beta_j\,\eta_j, \qquad \underline{\eta} = (\eta_1,\ldots,\eta_{2N})$$

$$|\hat{\Theta}_{\underline{\eta}}^R\rangle = \check{b}_1^{\times\,\eta_1}\check{b}_2^{\times\,\eta_2}\ldots\check{b}_{2N}^{\times\,\eta_{2N}}\,|\hat{\rho}_{NESS}\rangle, \qquad \langle\hat{\Theta}_{\underline{\eta}}^L| = \langle\hat{1}|\,\check{b}_1^{\eta_{2N}}\ldots\check{b}_2^{\eta_2}\check{b}_1^{\eta_1}, \tag{28}$$

where $\eta_j = \{0,1\}$. The left and right vectors satisfy the biorthogonality condition, $\langle\hat{\Theta}_{\underline{\eta}}^L|\hat{\Theta}_{\underline{\eta}}^R\rangle = 1$. Note that $\eta_j$ has the meaning of the occupation numbers of superoperators in the "energy" representation, whereas $\alpha_j$ entered in (13) corresponded to the occupation numbers in the "lattice" representation. For the left vacuum $\langle\hat{1}| \equiv \langle\hat{P}_{(0,0\ldots,0)}|$ we have $\langle\hat{1}|\check{\mathcal{L}} = 0$, therefore $\langle\hat{1}|\check{b}_l^\times = 0$. From the linear relation between $\check{b}_l^\times$ and $\underline{\check{c}}^+$, cf. Eq. (27), and from the properties $\langle\hat{P}_{(0,0\ldots,0)}|\check{c}_j^+ = 0$, it follows that the left vacuum of the Liouvillian $\check{\mathcal{L}}$ is the unit operator in the Fock-Liouville space. Similarly, the right vacuum is $|\hat{\rho}_G\rangle$, such that $\check{\mathcal{L}}|\hat{\rho}_G\rangle = 0$. It is fixed by the relations $\check{b}_l|\hat{\rho}_G\rangle = 0$. It corresponds to the vector with all $\eta_j = 0$. As can be seen from Eq. (28), this state describes a single NESS with $\lambda_{\underline{\eta}} = 0$ if all $\mathrm{Re}\,\beta_l > 0$.

However, if the spectrum has one or more kinetic modes with $\mathrm{Re}\,\beta_l = 0$, then the system implements a degenerate steady state (the so-called steady space) with $\lambda_{\underline{\eta}} = 0$. So, if there are dissipative modes with $\beta_l = 0$, as well as pairs of modes with $\beta_l + \bar{\beta}_{l'} = 0$, then their excitation changes the structure of the steady state density matrix as [38]:

$$|\hat{\rho}_{NESS}\rangle = |\hat{\rho}_G\rangle + \sum_{r:\,\beta_r=0}\alpha_r\check{b}_r^\times\,|\hat{\rho}_G\rangle + \sum_{(p\bar{p}):\,\beta_p+\beta_{\bar{p}}=0}(a_p/4)\,\check{b}_p^\times\check{b}_{\bar{p}}^\times\,|\hat{\rho}_G\rangle + \ldots,\ \alpha_r,\,a_p\in\mathbb{R}. \tag{29}$$

The unit coefficient in front of $|\hat{\rho}_{NESS}\rangle$ is fixed by the requirement of trace invariance, $\mathrm{Tr}\,\hat{\rho}_{NESS} = 1$, while the relations $\alpha_r = \alpha_r^*$ and $a_p = a_p^*$ are related with the requirement of the density matrix to be Hermitian, $\hat{\rho}_{NESS} = \hat{\rho}_{NESS}^+$. Therefore, the dark space is parametrized by the set of real numbers $\alpha_r,\,a_p,\ldots$ Dots in Eq. (29) denotes the terms which are constructed in a similar way: two pairs of modes with $\beta_l + \beta_{l'} = 0$ are excited, three pairs and so on.

The density matrix $|\hat{\rho}_G\rangle$ corresponding to the vacuum of kinetic modes ($\eta_1 = \ldots = \eta_{2N} = 0$) can be written in the Fock space in the Gaussian form [87]:

$$\hat{\rho}_G \sim \exp\left(-\frac{i}{2}\,\underline{\hat{w}}^T G\underline{\hat{w}}\right), \quad G = G^* = -G^T, \quad F = \tanh iG. \tag{30}$$

Such form of the density matrix guarantees the validity of the Wick theorem for the correlation functions in Majorana representation. The hermitian operator $\hat{\mathcal{H}} = (i/2)\,\underline{\hat{w}}^T G\underline{\hat{w}}$ involved in

---

[3]If the matrix $X$ is non-diagonalizable, the expression for the eigen values and eigen vectors of the Liouvillian becomes more involved, see Theorem 4.1 from Ref. [39].

Eq. (30) is interpreted as the effective Hamiltonian of the dissipative system in the non-equilibrium stationary state. We note that the matrix $F$ can be explicitly rewritten as the correlation matrix $F_{ij} = \mathrm{Tr}([\,\hat{w}_i\,,\,\hat{w}_j\,]\,\hat{\rho}_G)/2$. We note that information about ZKM do not appear in the spectrum of the matrix $G$ since it is absent in the matrix $F$.

### 3.4 Types and robustness of zero modes in closed and open systems

An important question concerns the similarities between the zero modes in the isolated system and the zero kinetic modes in the presence of dissipation. In other words, it is about the relation between the states that correspond to $\varepsilon_l = 0$ in the absence of dissipation and those with $\beta_l = 0$ in the presence of dissipation. To resolve this issue it is instructive to consider an auxiliary single-particle physical observable.

At first we recall how the zero mode state behaves in the isolated system. Let us consider the ground state $|0\rangle$ of the Hamiltonian (12) and the state $|1\rangle = \alpha_1^\dagger|0\rangle$, where $\varepsilon_1 = 0$, that has the same energy. Then for a single-particle operator $\hat{O} = i\sum_{jk} O_{jk}\hat{w}_j\hat{w}_k$ with a skew-symmetric matrix $O = -O^T$, we find

$$\delta\langle\hat{O}\,\rangle_{\mathrm{iso}} = \langle 1|\hat{O}|1\rangle - \langle 0|\hat{O}|0\rangle \equiv \mathrm{Tr}\left(\hat{O}\,\delta\hat{\rho}^{(2)}\right) = 4\,\underline{\chi}_1^T O\underline{\chi}_2. \tag{31}$$

Here $\delta\hat{\rho}^{(2)} = |1\rangle\langle 1| - |0\rangle\langle 0|$ is a change in the density matrix of the system when the zero mode is occupied. As we discussed above, we expect that $\delta\langle\hat{O}\,\rangle_{\mathrm{iso}}$ vanishes in the thermodynamic limit, $N \to \infty$, provided the matrix $O$ is local enough. Such a situation indicates that one cannot distinguish the states $|0\rangle\langle 0|$ and $|1\rangle\langle 1|$ by studying local single-particle observables. Such feature might serve as an indicator of topological phases in superconductors [91].

Now we turn back to the dissipative case. We start from transformation of Eq. (29) from the Fock-Liouville to the Fock space: $|\hat{\rho}_{NESS}\rangle \to \hat{\rho}_{NESS}$. Next, using Eqs. (15) and (27), we rewrite Eq. (29) as:

$$\hat{\rho}_{NESS} = \hat{\rho}_G + \hat{\rho}^{(1)} + \hat{\rho}^{(2)} + \dots \tag{32}$$

where

$$\hat{\rho}^{(1)} = -\frac{i}{2}\sum_{r:\,\beta_r=0}\alpha_r\sum_{l=1}^{2N} U_{lr}\hat{W}\{\hat{w}_l, \hat{\rho}_G\} \tag{33}$$

and

$$\hat{\rho}^{(2)} = \frac{1}{16}\sum_{(p\,\bar{p}):\,\beta_p+\beta_{\bar{p}}=0} a_p\sum_{l,l'=1}^{2N} U_{lp}U_{l'\bar{p}}[\hat{w}_l, \{\hat{w}_{l'}, \hat{\rho}_G\}]. \tag{34}$$

Using such density matrix we find for the single-particle operator

$$\delta\langle\hat{O}\,\rangle_{\mathrm{dis}}^{(1)} = \mathrm{Tr}\left(\hat{O}\,\hat{\rho}^{(1)}\right) \equiv 0 \tag{35}$$

and

$$\delta\langle\hat{O}\,\rangle_{\mathrm{dis}}^{(2)} = \mathrm{Tr}\left(\hat{O}\,\hat{\rho}^{(2)}\right) = \frac{1}{4}\sum_{(p\,\bar{p}):\,\beta_p+\beta_{\bar{p}}=0} a_p\,\mathrm{Im}\left(\underline{U}_p^T O\,\underline{U}_{\bar{p}}\right). \tag{36}$$

Here, the equality $\delta\langle\hat{O}\,\rangle_{\mathrm{dis}}^{(1)} = 0$ is due to the structure of the excitation of $\hat{\rho}^{(1)}$, cf. Eq. (33), in which the Gaussian density matrix is multiplied by an operator with a single Majorana operator $\hat{w}_l$. Thus, according to the Wick's theorem, such averages with an odd number of

$\hat{w}_l$ are zero. We note that the complete immunity to arbitrary operators represents a defining feature of systems hosting isolated Majorana state [53].

As the contribution of $\delta\langle\hat{\mathcal{O}}\rangle_{\text{dis}}^{(2)}$ is concerned, two cases for a pair $(p, \bar{p})$ are possible: (i) both kinetic modes have zero energy $\beta_p = \beta_{\bar{p}} = 0$; (ii) both modes are non-dissipative, $\text{Re}\,\beta_p = \text{Re}\,\beta_{\bar{p}} = 0$, but their energies are complex conjugated: $\beta_p = -\beta_{\bar{p}}$. In the case (i), the eigenvectors of $X$ belongs to the kernel of matrix $A$: $\underline{U}_{p(\bar{p})} \in \text{span}\{\underline{\chi}_1, \ldots, \underline{\chi}_{2N_M}\}$, see Eq. (66). Moreover, they can be chosen to be real. Then, the corresponding contribution in the right hand side of Eq. (36) vanishes identically, both for local and nonlocal operators $O$.

In the case (ii), there is the relation $\underline{U}_p = \underline{U}_{\bar{p}}^* \notin \ker A$, see Eq. (24), and filling modes with $\beta_p + \beta_{\bar{p}} = 0$ leads to a non-zero contribution

$$\delta\langle\hat{\mathcal{O}}\rangle_{\text{dis}}^{(2)} = -\frac{1}{2}\sum_p a_p\, \underline{\psi}_{2p-1}^T O \underline{\psi}_{2p} \tag{37}$$

We note that this result is similar to the result for the isolated system, see Eq. (31). Let us consider the operator $\hat{\mathcal{O}}^{(jk)} = [w_j, w_k]/(2i)$. The corresponding matrix $O^{(ij)}$ has the following structure: $O_{lm}^{(jk)} = (-1/2)(\delta_{jl}\delta_{km} - \delta_{jm}\delta_{kl})$. Then using Eq. (36), we find $\delta F_{jk} \equiv \delta\langle\hat{O}^{(jk)}\rangle_{\text{dis}}^{(2)} = (i/4)\sum_p a_p\, \text{Im}(\underline{U}_p\underline{U}_p^+)_{jk}$. We note that we interpret the correction $\delta\langle\hat{O}^{(jk)}\rangle_{\text{dis}}^{(2)}$ as the correction to the matrix $F$ since such $\delta F$ satisfies homogeneous Lyapunov-Sylvester equation, see (25) with $Y = 0$.

Thus, we argued that occupation of a single ZKM with $\beta_r = 0$ leads to an indistinguishable change in the structure of a degenerate non-equilibrium stationary state from the point of view of single-particle physical observables. This property of ZKM is different from the property of zero energy states in the isolated case described in Sec. 2. For the latter it holds only for local operators. Such a difference is probably explained by the fact that the NESS forms as a result of hybridization with the environment. In systems with the multiple ZKM, the hybridization behavior becomes more subtle (see the Sec. 4.3). Nevertheless, the NESS states remain indistinguishable to at least local observables. Therefore, a knowledge of the number of ZKM is crucial for fixing the dimensionality of the dark space in the presence of dissipation.

The feature of indistinguishability allows us to consider ZKM corresponding to exact zero eigen values of $X$, $\beta_r = 0$, as generalization of MM to the case of dissipative systems. As we will see below, it is convenient to split ZKM in two groups: the first one involves *robust* ZKM which existence is not related with the dissipative bath, the second group contains *weak* ZKM which exist due to peculiarities of hybridization between MM and the bath degrees of freedom.

## 3.5 Limit of vanishing dissipation

It is instructive to examine the transition to the dissipationless case. To construct this limit we introduce the dimensionless dissipation strength $\gamma$ and substitute the matrix $M$, see Eq. (22), by the matrix $\gamma M$. In the limit $\gamma \to 0^+$ we expect the spectrum and weights of kinetic excitations to exhibit the following asymptotic behavior:

$$(\beta_l, \beta_{\bar{l}}) \to \pm i\varepsilon_l, \qquad U \to \mathcal{W}R, \qquad R = \frac{1}{\sqrt{2}}\bigoplus_{l=1}^N \begin{pmatrix} 1 & i \\ i & 1 \end{pmatrix}_l. \tag{38}$$

This limiting case admits two interpretations. First, we may consider that pairs of kinetic modes with $\beta \to \pm i\varepsilon$ remain unpopulated in the steady state. The Gaussian NESS then

factorizes into a direct product of mixed states:

$$\hat{\rho}_G = \prod_{p=1}^{N} \left[ \frac{1}{2} \left( 1 + f_p \right) - f_p \, \hat{\alpha}_p^+ \, \hat{\alpha}_p \right] = \frac{1}{2^N} + \sum_p \frac{1}{2} f_p \left( 1 - 2\hat{\alpha}_p^+ \hat{\alpha}_p \right) + O(f_p^2), \tag{39}$$

where the occupation probability of the $p$-th Bogoliubov quasiparticle is $(1 - f_p)/2$. The amplitudes $0 \le f_p \le 1$ correspond to eigenvalues of the correlation matrix $F$, whose elements can be expressed as:

$$F_{ab} = -2i\gamma \sum_{j,k=1}^{2N} U_{aj} \frac{\left( U^{-1} M_i U^* \right)_{jk}}{\beta_j + \beta_k} \left( U^{-1} \right)_{kb}^*. \tag{40}$$

Alternatively, the $\gamma \to 0^+$ limit can be interpreted as a case where all kinetic mode pairs with $\beta \to \pm i\varepsilon$ become populated in the steady state. This requires spectral regularization $\beta \to \beta + \epsilon$, with $\lim_{\gamma \to 0} \gamma / \epsilon(\gamma) \to 0$. The regularized Lyapunov equation $X_\epsilon F + F X_\epsilon^T = Y$ then yields the trivial solution $F|_{\gamma \to 0} \to 0$ via (40), corresponding to a maximally entangled Gaussian state $\hat{\rho}_G^{(\epsilon)} = I_{2^N}/2^N$ that describes an uncorrelated fermionic ensemble. Kinetic mode occupations modify this density matrix according to Eq. (34):

$$\hat{\rho}_{NESS} = \hat{\rho}_G^{(\epsilon)} + \hat{\rho}^{(2)} + \ldots \xrightarrow{\gamma \to 0} \frac{1}{2^N} + \sum_p \frac{1}{2} a_p \left( 1 - 2\hat{\alpha}_p^+ \hat{\alpha}_p \right) + \ldots, \tag{41}$$

where we used Eqs. (10) and (24), and the relations $\hat{\rho}^{(1)} \to 0$, $\underline{\psi}_{2l-1(2l)} \to \underline{\chi}_{2l-1(2l)}$, $\sum_l U_{l\bar{p}} \hat{w}_l / 2 \to \hat{\alpha}_p^+$ in the limit $\gamma \to 0^+$. Enforcing consistency between Eqs. (39)-(41) yields [4]:

$$f_p = a_p = -i \sum_{a=1}^{N} R_{pa}^+ \frac{\left( \mathcal{W}^T M_i \mathcal{W} \right)_{a\bar{a}}}{\operatorname{Re} \tilde{\beta}_a} R_{\bar{a}p}, \qquad \operatorname{Re} \tilde{\beta}_a = \lim_{\gamma \to 0^+} \operatorname{Re} \beta_a. \tag{42}$$

This result demonstrates that under weak dissipation, the mode occupation probabilities are determined by the hybridization between the isolated system and dissipative fields (via matrix elements $(\mathcal{W}^T M_i \mathcal{W})_{a\bar{a}}$), and system spectral properties ($\operatorname{Re} \beta_a$). The thermal distribution $f_p \to \tanh(2\varepsilon_p/T)$ would indicate adiabatic convergence to the equilibrium Gibbs state at some temperature $T$. However, the general conditions for appearance of such equilibrium state for an arbitrary matrix $M$ remain non-trivial problem.

## 4 The condition for the existence of zero kinetic modes

### 4.1 The number of ZKM: algebraic approach

We start from the analysis of existence of ZKM, i.e. eigenmodes of the matrix $X$ with $\beta_l = 0$. It is convenient to present the matrix $M_r$ in the form of the Gram matrix

$$M_r = \ell \cdot \ell^T, \qquad \ell = [\underline{l}_1^r, \underline{l}_1^i, \ldots, \underline{l}_{N_B}^r, \underline{l}_{N_B}^i] \in \mathbb{R}^{2N \times 2N_B}. \tag{43}$$

---

[4]Note that in the limit $\gamma \to 0^+$, the matrices $F$ and $X$ have a common eigenbasis defined by the matrix $U$. This corresponds to the situation where the so-called spectral and purity gaps coincide [69]. For finite $\gamma$, however, this property generally no longer holds, and the two gaps characterize distinct features of the system.

Then using *Weinstein–Aronszajn identity* [92], we rewrite the characteristic polynomial of the matrix $X$ as

$$\mathcal{P}(\beta) = \det(\beta I_{2N} - X) = \det(A + \beta I_{2N}) \det(I_{2N_B} - \ell^T (A + \beta I)^{-1} \ell). \qquad (44)$$

We assume that there are $N_M$ zero energy modes, $\varepsilon_{1,\dots,N_M} = 0$, in the isolated system (with $N_M < N$). Next we use canonical representation of $A$, cf. Eq. (6), and rewrite the characteristic polynomial as

$$\mathcal{P}(\beta) = \beta^{2N_M} \prod_{l'=N_M+1}^{N} (\beta^2 + \varepsilon_{l'}^2) \det\left( I_{2N_B} + \sum_{l=1}^{N} \frac{\beta P^{(l)} - \varepsilon_l Q^{(l)}}{\beta^2 + \varepsilon_l^2} \right),$$

$$Q^{(l)} = \mathcal{C}_l^T i\sigma_y \, \mathcal{C}_l = -\left[ Q^{(l)} \right]^T \in \mathbb{R}^{2N_B}, \qquad P^{(l)} = \mathcal{C}_l^T \sigma_0 \, \mathcal{C}_l = \left[ P^{(l)} \right]^T \in \mathbb{R}^{2N_B}. \qquad (45)$$

Here $\sigma_0$ and $\sigma_y$ are standard Pauli matrices while the matrices $\mathcal{C}_l$ are given as

$$\mathcal{C}_l = \left[ \mathcal{C}_l^{(1)}, \dots, \mathcal{C}_l^{(N_B)} \right] \in \mathbb{R}^{2 \times 2N_B}, \qquad \mathcal{C}_l^{(v)} = \begin{pmatrix} \underline{\chi}_{2l-1} \cdot \underline{l}_v^r & \underline{\chi}_{2l-1} \cdot \underline{l}_v^i \\ \underline{\chi}_{2l} \cdot \underline{l}_v^r & \underline{\chi}_{2l} \cdot \underline{l}_v^i \end{pmatrix} \in \mathbb{R}^{2 \times 2}. \qquad (46)$$

We note that matrices $\mathcal{C}_l^{(v)}$ describe a hybridization of eigen functions of zero mode in the isolated system with dissipative fields $\underline{l}_v^{r,i}$.

To single out the behavior of the characteristic polynomial in the limit $\beta \to 0$, we rewrite Eq. (45) as

$$\mathcal{P}(\beta) = \beta^{2(N_M - N_B)} \prod_{l'=N_M+1}^{N} (\beta^2 + \varepsilon_{l'}^2) \det\left( \mathcal{Z}_0 + \beta \, \mathcal{Z}_1(\beta) + \beta^2 \, \mathcal{Z}_2(\beta) \right), \qquad (47)$$

where the matrices $\mathcal{Z}_{0,1,2}$ read

$$\mathcal{Z}_0 = \sum_{l=1}^{N_M} \mathcal{C}_l^T \mathcal{C}_l, \qquad \mathcal{Z}_1(\beta) = I_{2N_B} - \sum_{l'=N_M+1}^{N} \frac{\varepsilon_{l'} \, Q^{(l')}}{\beta^2 + \varepsilon_{l'}^2}, \qquad \mathcal{Z}_2(\beta) = \sum_{l'=N_M+1}^{N} \frac{P^{(l')}}{\beta^2 + \varepsilon_{l'}^2}. \qquad (48)$$

It is worth noticing that $\mathcal{Z}_{1,2}(\beta)$ is finite in the limit $\beta \to 0$. Let us assume that the matrix $\mathcal{Z}_0 = \mathcal{Z}_0^T$ has $s$ zero eigen values. Making transformation to the eigen basis of $\mathcal{Z}_0$,

$$Z_0 \to \mathcal{Z}_0 = R \, \Lambda \, R^T, \qquad \mathcal{Z}_{1,2} \to \tilde{\mathcal{Z}}_{1,2} = R \, \mathcal{Z}_{1,2} \, R^T,$$

$$\Lambda = \mathrm{diag}(\underbrace{0, \dots, 0}_{s}, \lambda_{s+1}, \lambda_{s+2}, \dots, \lambda_{2N_B}), \qquad (49)$$

we find for $\beta \to 0$:

$$\det\left( \mathcal{Z}_0 + \beta \mathcal{Z}_1(\beta) + \beta^2 \mathcal{Z}_2(\beta) \right) \simeq \begin{vmatrix} \beta \, \tilde{\mathcal{Z}}_1^{[1,1]} & \beta \, \tilde{\mathcal{Z}}_1^{[1,2]} \\ \beta \tilde{\mathcal{Z}}_1^{[2,1]} & \Lambda^{[2,2]} + \beta \, \tilde{\mathcal{Z}}_1^{[2,2]} \end{vmatrix} = \det\left( \beta \, \tilde{\mathcal{Z}}_1^{[1,1]} \right)$$

$$\det\left( \Lambda^{[2,2]} + \beta \left( \tilde{\mathcal{Z}}_1^{[2,2]} - \tilde{\mathcal{Z}}_1^{[2,1]} \left( \tilde{\mathcal{Z}}_1^{[1,1]} \right)^{-1} \tilde{\mathcal{Z}}_1^{[1,2]} \right) \right) = \beta^s \det \tilde{\mathcal{Z}}_1^{[1,1]} \det \Lambda^{[2,2]}. \qquad (50)$$

Here notations $[a, b]$ denote the corresponding blocks of the matrices involved. In particular, blocks $[1, 1]$ and $[2, 2]$ have $s \times s$ and $(2N_B - s) \times (2N_B - s)$ sizes, respectively. As one can

see from Eq. (4.1), the behavior of determinant in Eq. (47) for $\beta \to 0$ are determined by the number of zero eigen values of $\mathcal{Z}_0$, $s = \dim \ker \mathcal{Z}_0 = 2N_B - \text{rk}\,\mathcal{Z}_0$. Substituting Eq. (4.1) into Eq. (47), we obtain

$$\mathcal{P}(\beta \to 0) \sim \beta^{2N_M - \text{rk}\,\mathcal{Z}_0}, \qquad \text{rk}\,\mathcal{Z}_0 \leq 2\min(N_B, N_M). \tag{51}$$

It is convenient to use the following properties of the rank of the matrix:

$$\text{rk}\,\mathcal{Z}_0 = \text{rk}\left(\sum_{l=1}^{N_M} \mathcal{C}_l^T \mathcal{C}_l\right) = \text{rk}\left(\mathcal{B}^T \mathcal{B}\right) = \text{rk}\,\mathcal{B}, \tag{52}$$

where we introduce

$$\mathcal{B} = \begin{pmatrix} \mathcal{C}_1^{(1)} & \cdots & \mathcal{C}_1^{(N_B)} \\ \vdots & \ddots & \vdots \\ \mathcal{C}_{N_M}^{(1)} & \cdots & \mathcal{C}_{N_M}^{N_B} \end{pmatrix} = \tilde{\mathcal{W}}^T \cdot \ell \in \mathbb{R}^{2N_M \times 2N_B} \tag{53}$$

Here $\tilde{\mathcal{W}} = [\underline{\chi}_1, \underline{\chi}_2, \ldots, \underline{\chi}_{2N_M-1}, \underline{\chi}_{2N_M}] \in \mathbb{R}^{2N \times 2N_M}$ is nothing but the reduced $\mathcal{W}$, cf. Eq. (7), composed out of $2N_M$ first columns, corresponding to the zero modes. We note that the matrix $\mathcal{B}$ has the form of cross Gram matrix built out of two sets of vectors $\underline{\chi}_a$, $a = 1, \ldots, 2N_M$, and $l_v^{r,i}$, $v = 1, \ldots, N_B$. The matrix $\mathcal{B}$ has a transparent meaning. The vectors $\chi_a$, $a = 1, \ldots, 2N$, form a basis in $\mathbb{R}^{2N}$. The eigen vectors $\chi_a$, $a = 1, \ldots, 2N_M$, correspond to the zero mode subspace in $\mathbb{R}^{2N}$. The elements of the matrix $\mathcal{B}$ encodes the part of the dissipative field belonging to the zero mode subspace: $\ell = \mathcal{B}^T \tilde{\mathcal{W}} + \ell_\perp$.

For $N_M > N_B$ ($N_M < N_B$), it is more practical to compute $\text{rk}\,\mathcal{Z}_0$ from the columns (rows) of the matrix $\mathcal{B}$. Therefore, we find that the number of ZKM is given as

$$N_0 = 2N_M - \text{rk}\,\mathcal{B}, \qquad \text{rk}\,\mathcal{B} \leqslant 2\min(N_B, N_M). \tag{54}$$

In the case $N_M > N_B$, the number of ZKM is bounded from below as

$$N_0 \geqslant 2(N_M - N_B) \tag{55}$$

The equality corresponds to the condition $\text{rk}\,\mathcal{B} = 2N_B$ and the following behavior of the characteristic polynomial $\mathcal{P}(\beta \to 0) \sim \beta^{2(N_M - N_B)}$. In this case, we term such ZKM as *robust zeroes*. In the opposite case $N_M \leqslant N_B$, the robust ZKM are absent.

However, one can engineer additional $r$ zeroes, $N_0 = 2(N_M - N_B) + r$, due to reduction of the rank of the matrix $\mathcal{B}$: $\text{rk}\,\mathcal{B} = 2\min(N_B, N_M) - r$. We will term such additional zeroes as *weak zeroes*. The straight forward way to maximize the number of ZKM is to engineer the dissipative fields that do not belong to the zero mode subspace of eigen vectors of isolated system. In this way the matrix $\mathcal{B}$ is identical zero such that the dissipation does not reduce the number of zero modes $N_0 = 2N_M$ (it can be considered as generalization of the construction proposed in Refs. [68–70] for the Liouvillians without unitary evolution).

## 4.2 The number of ZKM: operator approach and symmetries

The results of the previous section for the number of ZKM can be explained in terms of the operator approach and symmetries of the Liouvillian. We remind that $\hat{\alpha}_a = \left(\underline{\chi}_{2a-1} + i\underline{\chi}_{2a}\right) \cdot \underline{\hat{w}}$

and $\hat{L}_v = \left( \underline{l}_v^r + i \underline{l}_v^i \right) \cdot \underline{\hat{w}}$. Then making elementary transformations of the matrix $\mathcal{B}$, see Eq. (53), we find

$$\mathrm{rk}\,\mathcal{B} \,=\, \mathrm{rk}\,\mathcal{D}\,, \quad \mathcal{D} = \begin{pmatrix} \mathcal{G}_1^{(1)} & \cdots & \mathcal{G}_1^{(N_B)} \\ \vdots & \ddots & \vdots \\ \mathcal{G}_{N_M}^{(1)} & \cdots & \mathcal{G}_{N_M}^{(N_B)} \end{pmatrix}\,, \quad \mathcal{G}_a^{(v)} = \begin{pmatrix} |\,\hat{\alpha}_a^+,\,\hat{L}_v\,| & |\,\hat{\alpha}_a^+,\,\hat{L}_v^+\,| \\ |\,\hat{\alpha}_a,\,\hat{L}_v\,| & |\,\hat{\alpha}_a,\,\hat{L}_v^+\,| \end{pmatrix}. \quad (56)$$

Here we introduce the following scaling product of two operators: $|\,\hat{A},\,\hat{B}\,| = 2^{-N} \mathrm{Tr}\{\hat{A}^+,\hat{B}\}$ [5]. We note that the matrix $\mathcal{D}$ involves projections of jump operators to the operators corresponding to the eigen modes of the isolated system. Next, we find

$$N_0 = 2N_M - \mathrm{rk}\,\mathcal{D} = \dim\ker\mathcal{D}^T = \underbrace{-\,\mathrm{ind}\,\mathcal{D}}_{\text{robust ZKMs}} + \underbrace{\dim\ker\mathcal{D}}_{\text{weak ZKMs}}. \quad (57)$$

Therefore, the number of the robust ZKM is determined by the index of the matrix $\mathcal{D}$, $\mathrm{ind}\,\mathcal{D} = 2(N_B - N_M)$ [93], while the number of the weak ZKM coincides with dimension of its kernel, $r = \dim\ker\mathcal{D}$.

As a next step, we mention that a creation operator for the ZKM in the Fock-Liouville space, $\check{b}_r^\times$, see Eq. (29), commute with the Lindbladian (27). Also $\check{b}_r^\times$ changes fermionic parity of the system. In the Fock space the above mentioned properties can be written as:

$$\left[\,\hat{\mathcal{L}},\,\hat{b}_r\,\right] = 0\,, \quad \{\,\hat{W},\,\hat{b}_r\,\} = 0\,. \quad (58)$$

We note that after the substitution $\hat{\mathcal{L}} \to \hat{H}$ Eq. (58) coincides with definition of the so-called strong zero modes in the isolated systems with unitary dynamics [94–99].

Definitions (58) for the ZKM operators and invariant expressions (56)-(57) for their number can be used to construct ZKM in systems with more involved dissipation than the one considered in the present manuscript. Also definitions (58) demonstrates that the ZKM operators realize the weak symmetries of the Lindbladian [47–49, 51].

Interestingly, the ZKM can be related with the strong symmetries of the Lindbladian as well. Let us construct the following unitary operator:

$$\hat{S}(\underline{\theta}) = \exp\left( \sum_{m,m'=1}^{2N_M} D_{mm'}(\underline{\theta})\,\hat{b}_m\,\hat{b}_{m'} \right)\,, \qquad [\,\hat{H},\,\hat{S}(\underline{\theta})\,] = 0\,, \qquad [\,\hat{L}_v,\,\hat{S}(\underline{\theta})\,] = 0\,, \quad (59)$$

which correspond to rotations in the Fock space of $N_M$ zero modes. Here operators $\hat{b}_{2a-1} = \hat{b}_a'$ and $\hat{b}_{2a} = \hat{b}_a''$, see Eq.(10), correspond to the modes of the isolated system with zero energy, $\varepsilon_m = 0$. Obviously, such operator $S$ commutes with the Hamiltonian $[\,\hat{H},\,\hat{S}\,] = 0$. We choose the matrix $D$ to be real and skew-symmetric: $D \in \mathbb{R}^{2N_M \times 2N_M}$, $D = -D^T$. In order to satisfy relations $[\,\hat{L}_v,\,\hat{S}\,] = 0$ the matrix $S$ should belong to the kernel of the matrix $\mathcal{B}^T$: $D \in \ker\mathcal{B}^T = \mathrm{span}\{\underline{y}_1, \underline{y}_2, \ldots, \underline{y}_{N_0}\}$, where $\mathcal{B}^T \underline{y}_i = 0$. In this case, the matrix $D$ can be constructed explicitly with the help of $N_0(N_0 - 1)/2$ real parameters $\theta_{ij}$ as follows

$$D(\underline{\theta}) = \sum_{i<j=1}^{N_0} \theta_{ij} \left( \underline{y}_i\,\underline{y}_j^T - \underline{y}_j\,\underline{y}_i^T \right)\,, \qquad \theta_{ij} \in \mathbb{R}. \quad (60)$$

---

[5] We note that such scalar product is different from the one in the form of Hilbert-Schmidt, (14), due to the presence of anticommutator. Still it satisfies the basic properties of scalar products: i) $|\,(a\hat{A} + b\hat{B})\,,\,\hat{C}\,| = a|\,\hat{A},\,\hat{C}\,| + b|\,\hat{B},\,\hat{C}\,|$, ii) $|\,\hat{A},\,\hat{B}\,| = |\,\hat{B},\,\hat{A}\,|^*$, iii) $|\,\hat{A},\,\hat{A}\,| \geq 0$ и $|\,\hat{A},\,\hat{A}\,| = 0 \Leftrightarrow A = 0$.

In exceptional cases, $N_0 = 0$ ($\mathrm{rk}\,\mathcal{B} = 2N_M$) and $N_0 = 1$ ($\mathrm{rk}\,\mathcal{B} = 2N_M - 1$) the unitary operators in the form of Eq. (59) do not exist. In the case $1 < N_0 < 2N_M$ the operators $\hat{S}(\underline{\theta})$ do exist and are parametrized by $N_0(N_0 - 1)/2$ real parameters. In the absence of dissipation, the dimensionality of the parameter space becomes $2N_M(2N_M - 1)/2$ as expected.

To determine the number of distinct eigenvalues of the operators (59), we note that the basis vectors of $\ker \mathcal{B}^T$ can be chosen to be real and orthogonal: $\Phi = \{\underline{y}_1, \underline{y}_2, \ldots, \underline{y}_{N_0}\} \in \mathbb{R}^{2N_M \times N_0}$ with $\Phi^T \Phi = I_{N_0}$. Introducing a set of Majorana operators

$$\hat{\omega}_l = \underline{y}_l \cdot \hat{\underline{b}}, \qquad \{\hat{\omega}_l, \, \hat{\omega}_m\} = 2\delta_{lm}, \qquad l, m = 1, \ldots, N_0 \tag{61}$$

the anti-Hermitian operator appearing in (59) can be expressed as:

$$\sum_{m,m'=1}^{2N_M} D_{mm'}(\underline{\theta})\, \hat{b}_m\, \hat{b}_{m'} = \hat{\underline{\omega}} \cdot \Theta\, \hat{\underline{\omega}}, \qquad \Theta_{lm} = \mathrm{sign}(m - l)\,\theta_{lm}. \tag{62}$$

The real antisymmetric matrix $\Theta$ can be brought to canonical form $\Theta_c = \mathcal{Q}^T \Theta \mathcal{Q}$, which differs for even and odd $N_0$:

$$\Theta_c^{(ev)} = \bigoplus_{j=1}^{N_0/2} \begin{pmatrix} 0 & i\kappa_j \\ -i\kappa_j & 0 \end{pmatrix}, \; N_0 \in 2\mathbb{Z}_+; \quad \Theta_c^{(od)} = \{0\} \bigoplus_{j=1}^{(N_0-1)/2} \begin{pmatrix} 0 & i\kappa_j \\ -i\kappa_j & 0 \end{pmatrix}, \; N_0 \in 2\mathbb{Z}_+ + 1. \tag{63}$$

Consequently, the minimal Fock space dimension for the operator $\hat{S}(\underline{\theta})$ is $2^{\lfloor N_0/2 \rfloor}$, where the floor brackets denote rounding down to the nearest integer. This space can be constructed by introducing fermionic operators: $\hat{\gamma}_l = (\mathcal{Q}_{2l-1} + i\mathcal{Q}_{2l}) \cdot \hat{\underline{\omega}}$. In terms of these fermions, the unitary operator (59) and its spectrum are given by:

$$\hat{S}(\underline{\theta}) \sim \exp\left( 2 \sum_{l=1}^{\lfloor N_0/2 \rfloor} \kappa_l(\underline{\theta})\, \hat{\gamma}_l^+ \hat{\gamma}_l \right), \qquad \Lambda_{\{s\}} = 2 \sum_{l=1}^{\lfloor N_0/2 \rfloor} s_l\, \kappa_l(\underline{\theta}), \quad s_l = 0, 1. \tag{64}$$

Thus, in the general case (arbitrary $\underline{\theta}$), such operators possess $2^{\lfloor N_0/2 \rfloor}$ distinct eigenvalues. In accordance with the Theorem A.1 of Ref. [47] this fact guarantees the existence of $2^{\lfloor N_0/2 \rfloor}$ NESS at least.

## 4.3   The number of ZKM: the structure of eigen vectors

Let us now discuss the structure of the (right) eigen vectors of the matrix $X$, corresponding to ZKM,

$$X\underline{U}_a = 0, \qquad a = 1, \ldots N_0 = 2N_M - \mathrm{rk}\,\mathcal{B}. \tag{65}$$

Let us demonstrate that such eigen vectors belong to the kernel of the matrix $A$ and can be constructed as follows

$$\underline{U}_a = \sum_{b=1}^{2N_M} y_{a,b} \underline{\chi}_b = \tilde{\mathcal{W}} \underline{y}_a, \qquad \underline{\chi}_b \in \ker A. \tag{66}$$

where $\underline{y}_a$ are the vectors from the kernel of the matrix $\mathcal{B}^T$, i.e. $\mathcal{B}^T \underline{y}_a = 0$, $a = 1, \ldots, N_0$. Indeed, substituting the ansatz (66) into Eq. (65), we find the following matrix equation

$$\ell^T \mathcal{B}^T \underline{y}_a = 0, \tag{67}$$

that is trivially satisfied due to $\underline{y}_a \in \ker \mathcal{B}^T$. Since in Eq. (66) we construct eigen vectors corresponding to all existing ZKM, we can state that there is no other zero eigen vectors of $X$.

The result (66) allows us to put the following physical picture of the effect of dissipation on the MM. Below we assume that $N_B < N_M$. The first effect is destruction of the $2N_M - N_0 = \text{rk}\,\mathcal{B}$ number of MM: the corresponding eigen values $\beta_a$ becomes nonzero. For the remaining $N_0$ ZKM with $\beta_a = 0$, their eigen vectors become linear combination of the eigen vectors of MM in the isolated system. This fact can be naturally interpreted as hybridization between MM states due to interaction with the bath.

We note the subtlety of the transformation of the eigenvectors of $X$ in the limit of vanishing dissipation, see Sec. 3.5. In the presence of dissipation the eigenvectors of the matrix $X$ for the ZKM not only belongs to the kernel of $A$ but also are real. In the isolated case the eigenvectors of the matrix $A$ for MM are conveniently chosen as complex vectors, $\underline{\chi}_{2a-1} \pm i\underline{\chi}_{2a}$, to be consistent with the structure of eigenvectors for states with nonzero energies $\varepsilon_b > 0$. We note that for the MM in the isolated system we could choose a real eigenvectors of the matrix $A$. Therefore, we will assume the transformation of $U \to \mathcal{W}R$ in the limit of vanishing dissipation (having in mind subtlety discussed above).

It is worthwhile to mention that the structure of the ZKM eigen vectors is similar to the one of the unperturbed states at Landau level in two-dimensional electron gas with perpendicular magnetic field in the presence of impurities with $\delta$-function potential. As known [100], if the number of impurities $n_{\text{imp}}$ is smaller than the number of states at the Landau level $n_L$, then there are $n_L - n_{\text{imp}}$ states those wave functions are constructed as linear combinations of unperturbed Landau level wave functions.

## 4.4  Recipes to generate weak ZKM

The results above provide the exact expression for the number of robust ZKM: $\max\{2(N_M - N_B), 0\}$. Analysis of the expression (52)-(53) allows us to give several practical recipes for inducing weak ZKM. As we discussed above, if one knows the eigen functions of the zero modes in the isolated system then it is possible to construct dissipative field not belonging to the zero mode subspace such that the total number of ZKM coincides with that in the absence of dissipation. If only some of the dissipative fields are orthogonal to the zero mode eigen functions then we are able to have weak ZKM those number lies between $2N_M$ and $2(N_M - N_B)$ (in the case $N_M > N_B$). We list below several ways to achieve weak ZKM.

(a) *Linear dependence of dissipative fields.*
From practical point of view, more interesting is construction of the weak ZKM without explicit knowledge of eigenvectors of the isolated Hamiltonian. In this case, in order to produce weak ZKM the dissipative fields should be linear dependent. For example, the linear dependence is realized inside a single bath, $\underline{l}_v^r \sim \underline{l}_v^i$. It corresponds to the following relation between the rates of creation and annihilation of fermions in the jump operator $L_v$: $\mu_{v,l} = e^{i\phi}\nu_{v,l}^*$. Such a linear dependent dissipative field reduces the number of zero modes by 1 rather than by 2. An important example, of such a situation is the self-adjoint jump operator, $L_v = L_v^+$, that corresponds to $\underline{l}_v^i \equiv 0$. We note that such a condition for creating weak ZKM was indicated in Ref. [70]. Note that the situation with self-adjoint Lindblad operators is naturally realized when the system is influenced by a set of projective measurements, which in recent years has been actively studied in

the literature [19, 81–84, 86]. Importantly, the linear dependence of dissipative field is not necessary restricted to a single bath. Generically, one can construct weak ZKM by creating linearly dependent dissipating fields from different baths, $v \neq v'$. We are not aware of discussion of such proposal before.

(b) *Dissipative fields acting in the subspace of gapped eigen states.*
Some dissipative field $l_v^r \notin \ker A$ (or $l_v^i \notin \ker A$), i.e. it belongs to linear hull of wave functions $\{\underline{\chi}_{m'}\}$ corresponding to the nonzero eigen values of $A$, $\varepsilon_{m'} > 0$. Since the wave functions are orthogonal, $\underline{\chi}_n \cdot \underline{\chi}_m = \delta_{n,m}$, the corresponding column of the matrix $\mathcal{B}$ vanishes. Each such dissipative field produces one weak ZKM. Such mechanism of generation of weak ZKM generalizes ideas proposed in Ref. [68–70] to the case of the presence of unitary evolution in Lindbladian.

(b′) *Orthogonality to gapless modes*
Each of $2N_B$ dissipative fields is orthogonal to a wave function of MM of the isolated system: $\underline{\chi}_{2l} \cdot \underline{l}_v^{r,i} = 0$ or $\underline{\chi}_{2l-1} \cdot \underline{l}_v^{r,i} = 0$ for $v = 1, \ldots, N_B$ and $\varepsilon_l = 0$. The number of weak ZKM, $r$, is equal to the dimension of the linear hull of such MM wave functions to which all dissipative fields are orthogonal.

(c) *Spatially structured dissipative fields*
Let us assume that the wave functions $\chi_a$ of the MM of the isolated system are spatially structured: $\underline{\chi}_1 \in \mathcal{R}_1, \ldots, \underline{\chi}_{2N_M} \in \mathcal{R}_{2N_M}$. If the dissipative field $\underline{l}_v^{r(i)}$ acting on the spatial region $\bar{\mathcal{R}}$, does not affect the spatial regions supporting the MM wave functions, $\bar{\mathcal{R}} \cap (\mathcal{R}_1 \cup \ldots \cup \mathcal{R}_{2N_M}) = \varnothing$, then the overlaps $\underline{\chi}_a \cdot \underline{l}_v^{r(i)}$ are zero. Therefore, each such spatially structured dissipative field produces one weak ZKM.

(c′) *Spatially localized dissipative fields*
Let us assume that each dissipative field is spatially structured and $\underline{l}_1^{r,i}, \ldots, \underline{l}_{N_B}^{r,i} \in \bar{\mathcal{R}}$. Also we assume that they do not affect the region $\mathcal{R}_a$ where a wave function $\underline{\chi}_a$ of the MM is localized. There exist $r$ ZKM provided $\bar{\mathcal{R}} \cap (\mathcal{R}_1 \cup \ldots \cup \mathcal{R}_r) = \varnothing$.

It is worthwhile to mention that the cases (c) and (c′) are quite natural when applying wide-band leads to the TS. Indeed, since such leads are modeled by local dissipative fields $\underline{t}^{r,i}$ [101], the spatial region of their effect on the TS may not overlap with the regions in which MM are localized. In this case, weak ZKM can be induced in the system, which must be taken into account when analyzing the current flow through the TS [101].

Below we will illustrate the above cases on the particular examples of the dissipative fields acting on the generalized Kitaev chain.

# 5 Generalized Kitaev chain

## 5.1 Formulation of model in the BDI class

Let us illustrate general results discussed in the previous sections with the help of a particular model of the topological superconductor. We will study the generalized Kitaev

chain [102–108] described by the one-dimensional Hamiltonian of the spin-polarized fermions:

$$\hat{H} = \sum_{j=1}^{N}(\epsilon_j - \mu)\,\hat{c}_j^+ \hat{c}_j + \sum_{r=1}^{N_r}\sum_{j=1}^{N-r}\left(\,t_r\,\hat{c}_j^+ \hat{c}_{j+r} + \Delta_r\,\hat{c}_j\,\hat{c}_{j+r} + h.c.\right). \tag{68}$$

We assume open boundary conditions. Here $\epsilon_j$ stands for the on-site energies and $\mu$ denotes the chemical potential. The parameters $t_r$ and $\Delta_r$ determine the hopping and pairing amplitudes, respectively. We emphasize that the hopping and pairing are active between sites $j$ and $j+r$ with $r = 1, \ldots, N_r$ only. The standard Kitaev model [53] corresponds to the case $N_r = 1$.

The Hamiltonian (68) has Bogoliubov-de Gennes symmetry. In order to elucidate it, we rewrite the Hamiltonian (68) as

$$\hat{H} = \frac{1}{2}\sum_{j,j'}\begin{pmatrix}\hat{c}_j^+ & \hat{c}_j\end{pmatrix}\mathcal{H}_{jj'}\begin{pmatrix}\hat{c}_{j'}^+ \\ \hat{c}_{j'}\end{pmatrix}, \qquad \mathcal{H}_{jj'} = \begin{pmatrix}h_{jj'} & -\Delta_{jj'}^* \\ \Delta_{jj'} & -h_{jj'}^*\end{pmatrix}, \qquad \tau_x \mathcal{H}^T \tau_x = -\mathcal{H}. \tag{69}$$

Here we introduce the following matrices

$$h_{jj'} = (\epsilon_j - \mu)\delta_{jj'} + \sum_{r=1}^{N_r}(t_r\delta_{j',j+r} + t_r^*\delta_{j',j-r}), \qquad \Delta_{jj'} = \sum_{r=1}^{N_r}\Delta_r(\delta_{j',j+r} - \delta_{j',j-r}). \tag{70}$$

The matrix $\tau_x$ is the Pauli matrix acting in the particle-hole space. In the case of real parameters $t_r = t_r^*$ and $\Delta_r = \Delta_r^*$ the Hamiltonian (69) has time-reversal symmetry, $\mathcal{H}^* = \mathcal{H}$. In this case, the Hamiltonian (68) belongs to the BDI symmetry class in Altland-Zirnbauer classification [109, 110]. This symmetry class has topological phases in one spatial dimension classified by the integer numbers $N_M \in \mathbb{Z}$ [111–113].

Below we consider the case $\epsilon_j \equiv 0$. Then it is convenient to rewrite the Hamiltonian (68) in the Majorana representation (3):

$$\hat{H} = -\frac{i}{2}\Big[\,\sum_{j=1}^{N}(-\mu)\,\hat{w}_{2j-1}\hat{w}_{2j} - \sum_{r=1}^{N_r}\sum_{j=1}^{N-r}\Big((\Delta_r - t_r)\,\hat{w}_{2j-1}\hat{w}_{2j+2r} + (\Delta_r + t_r)\,\hat{w}_{2j}\,\hat{w}_{2j+2r-1}\Big)\Big]. \tag{71}$$

In Fig. 5.1a we illustrate how Majorana operators in the Hamiltonian (71) are connected in the case $N_r = 2$. We note that there is no links between Majorana operators at sites $j$ and $j'$ with the same parity. It is the consequence of the time reversal symmetry of the Hamiltonian (71) that is realized by the anti-unitary operator $\hat{\mathcal{T}}$ as $\hat{\mathcal{T}}\,\hat{H}\,\hat{\mathcal{T}}^{-1} = \hat{H}$. The anti-unitary operator has the following properties:

$$\hat{\mathcal{T}}\,\hat{w}_j\,\hat{\mathcal{T}}^{-1} = (-1)^{j-1}\hat{w}_j, \qquad \hat{\mathcal{T}}\,i\,\hat{\mathcal{T}}^{-1} = -i. \tag{72}$$

The block structure of the matrix $A$ corresponding to the Hamiltonian (71) implies that the vectors $\underline{\chi}_a$ with odd and even $a$ lives on odd and even sites, respectively:

$$\underline{\chi}_{2a,2l-1} = \underline{\chi}_{2a-1,2l} \equiv 0. \tag{73}$$

According Eq. (11), the relations (73) implies that the parameters $\hat{u}_a$ and $\hat{v}_a$ of the Bogoliubov transformation are real.

The simplest model of the type (71) in which one can realize $2N_M$ MM corresponds to the following choice of parameters: $N_r = N_M$, $\mu = 0$, $t_r = \Delta_r = t\,\delta_{r,N_M} \in \mathbb{R}$. In what follows we

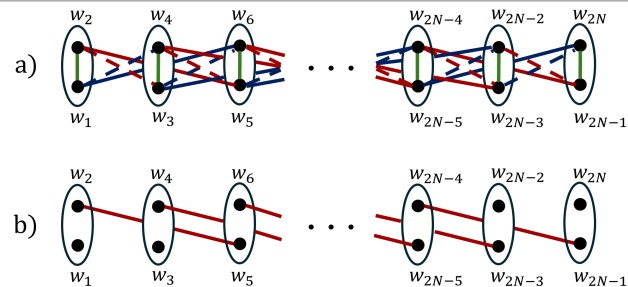

Figure 1: Schematic illustration of different terms in the generalized Kitaev chain (71) for $N_r = 2$: (a) general situation of nonzero $t_r$ and $\Delta_r$ for $r = 1, 2$, (b) the symmetric point with $\mu = t_1 = \Delta_1 = 0$ и $t_2 = \Delta_2 = t$. In the later case there are zero energy localized excitations corresponding to the Majorana operators $w_1, w_3$, $w_{2N-2}$, and $w_{2N}$.

term such choice of parameters as *symmetric points* of the system. At the symmetric points the Hamiltonian (71) becomes

$$\hat{H} \to \hat{H}' = i\,t \sum_{j=1}^{N-N_M} \hat{w}_{2j}\, \hat{w}_{2j+2N_M-1}\,. \tag{74}$$

We emphasize that exactly $2N_M$ Majorana operators ($w_1$, $w_3$,..., $w_{2N_M-1}$ on the left part of the chain and $w_{2N-2N_M},\ldots,w_{2N-2}, w_{2N}$ on the right part of the chain) are not involved in the Hamiltonian (74), see Fig. 5.1b. Therefore, such Majorana operators do commute with the Hamiltonian (74) and realize $2N_M$ MM. We mention that a generic Hamiltonian (71) can be adiabatically transformed to the Hamiltonian (74) without closing the bulk gap. We note that the Hamiltonian (74) corresponds to the matrix $A$ in the canonical form, see Eq. (6). Therefore, its eigen vectors are localized at a single site: $\chi_{a,j} = \delta_{ja}$.

## 5.2 The case $N_B = N_M = 1$: a single Majorana pair and single dissipative bath

Let us illustrate general theoretical results derived above in the simplest nontrivial case of $N_M = N_B = 1$. According to Eq. (57), there is no robust ZKM. In order weak ZKM to exist the rank of the matrix $\mathcal{B} \in \mathbb{R}^{2\times2}$ has to be reduced, $\mathrm{rk}\,\mathcal{B} < 2$. A single weak ZKM, $N_0 = 1$, is generated provided $\mathrm{rk}\,\mathcal{B} = 1$. This condition is equivalent to the condition of zero determinant:

$$\det \mathcal{B} = \left(\underline{\chi}_1 \cdot \underline{l}^r\right)\left(\underline{\chi}_2 \cdot \underline{l}^i\right) - \left(\underline{\chi}_1 \cdot \underline{l}^i\right)\left(\underline{\chi}_2 \cdot \underline{l}^r\right) = 0. \tag{75}$$

Here $\underline{\chi}_1$ and $\underline{\chi}_2$ are MM wave functions in the isolated system. A single ZKM is realized in the following cases:

(a) The jump operators are Hermitian, $L_v = L_v^+$, then $\underline{l}^i \equiv 0$, and the matrix $\mathcal{B}$ reads

$$\mathcal{B} = \begin{pmatrix} \underline{\chi}_1 \cdot \underline{l}^r & 0 \\ \underline{\chi}_2 \cdot \underline{l}^r & 0 \end{pmatrix}, \quad \ker \mathcal{B}^T = \underline{y} \sim \begin{pmatrix} \underline{\chi}_2 \cdot \underline{l}^r \\ -\underline{\chi}_1 \cdot \underline{l}^r \end{pmatrix}, \quad \underline{U}_1 \sim (\underline{\chi}_2 \cdot \underline{l}^r)\underline{\chi}_1 - (\underline{\chi}_1 \cdot \underline{l}^r)\underline{\chi}_2\,. \tag{76}$$

The structure of the eigen-vector $\underline{U}_1$ demonstrates that the interaction with the dissipative bath results in the hybridization of the MM wave functions.

(b) The dissipative fields are in the linear hull, which is orthogonal to one of the MM wave functions, e.g. $\underline{l}^{r,i} \in \text{span}\{\underline{\chi}_2, \underline{\chi}_3, \ldots, \underline{\chi}_{2N_M}\}$, where $\varepsilon_1 = \varepsilon_2 = 0$ and $\varepsilon_3, \ldots, \varepsilon_{2N_M} \neq 0$. In this case, we find

$$\mathcal{B} = \begin{pmatrix} 0 & 0 \\ \underline{\chi}_2 \cdot \underline{l}^r & \underline{\chi}_2 \cdot \underline{l}^i \end{pmatrix}, \quad \ker \mathcal{B}^T = \underline{y} \sim \begin{pmatrix} 1 \\ 0 \end{pmatrix}, \quad \underline{U} \sim \underline{\chi}_1. \tag{77}$$

Since the dissipative fields do not act on the MM state $\underline{\chi}_1$, it survives in the presence of dissipation. In contrast, the other MM state, $\underline{\chi}_2$, is destroyed by the dissipation.

In order to induce two weak ZKM, the matrix $\mathcal{B}$ has to be identically zero. It implies that the dissipative fields are orthogonal to the MM wave functions. Therefore, two weak ZKM coincides with the MM of the isolated system, $\underline{U}_1 \sim \underline{\chi}_1$ and $\underline{U}_2 \sim \underline{\chi}_2$.

## 5.3   Results of numerical calculations for the case $N_M = N_B = 1$

To demonstrate the above-described features of the system in the case $N_M = N_B = 1$, we present the results of numerical calculations for the standard Kitaev chain, Eq. (68) with $N_r = 1$, and parameters realizing the nontrivial topological phase: $t_1 = t$, $\Delta_1 = 0.8t$, $\mu = 0.5t$. The spatial dependence of the dissipative fields $\underline{l}^r$ and $\underline{l}^i$ is assumed to be random, but the nature of their distribution varied within the framework of the mechanisms (a)-(c), Sec. 4.4, for generation of weak ZKM.

Let us first consider the case of an isolated system corresponding to the absence of dissipative fields, i.e. $\underline{l}^{r,i} \equiv 0$, see Fig.2a. In a non-trivial topological phase, a pair of MM is implemented in the system, the wave functions of which, $\chi_1$ and $\chi_2$, are localized at opposite edges of the chain, as shown in Fig.2b. In this case, the spectrum of the energies $\varepsilon_a$ is real ($\beta_a = i\varepsilon_a$), symmetric with respect to zero, with a pair of zeros corresponding to paired MM, see Fig. 2c. We note that for the chain of a finite length $L$, the Majorana state energy is generally non-zero, i.e. $\varepsilon = \underline{\chi}_1 \cdot A \underline{\chi}_2 \sim e^{-bL}$, see e.g. [108]. This splitting determines the characteristic time $\tau_{\text{coh}} \sim \hbar/|\varepsilon|$, during which a phase error occurs in an isolated Majorana qubit [53]. In our numerical calculations, which are performed for a finite $L$ obviously, we restricted consideration to regimes where $\varepsilon$ vanishes within machine precision, thus effectively setting $\varepsilon = 0$.

When connecting the system to an external bath described by self-adjoint Lindblad operators, $\hat{L} = \hat{L}^+$, one component of the dissipative field vanishes, $\underline{l}^i = 0$ (see the blue dotted line in Fig. 2d). Such case is an example of the situation described in item (a) in Sec. 4.4. Then the modes with $|\text{Im}\,\beta| > 0$ become attenuated (they acquire $\text{Re}\,\beta > 0$). The pair of MM transforms into a single weak ZKM and a mode with $\text{Re}\,\beta > 0$ but $\text{Im}\,\beta = 0$, see Fig. 2f. The spatial distribution of the weak ZKM (the right eigen vector of $X$) becomes a linear combination of the MM wave functions in the isolated system: $\underline{U} \sim (\underline{\chi}_2 \cdot \underline{l}^r)\,\underline{\chi}_1 - (\underline{\chi}_1 \cdot \underline{l}^r)\,\underline{\chi}_2$, see Eq. (76) and Fig. 2e. As a result, the weak ZKM is localized near both edges of the chain. The partner state corresponding to the weak ZKM has $\text{Re}\,\beta > 0$ and $\text{Im}\,\beta = 0$, see Fig.2f. This state does not belong to $\ker A$ and, thus, spreads over the whole chain, see the dashed curve in Fig.2e.

The third line of Fig. 2 corresponds to the mechanism of generation of weak ZKM, in which the dissipative fields are orthogonal to one of the Majorana modes, e.g., $\underline{l}^{r,i} \cdot \underline{\chi}_1 = 0$ (see item (b) in Sec. 4.4). In Fig. 2g the dissipative fields were chosen as $\underline{l}^{r,i} = \sum_{s=2}^{6} c_s^{r,i} \underline{\chi}_s$ with random coefficients $c^{r,i} \in [0, 1]$. As can be seen from Fig. 2g and 2h, the spatial structure

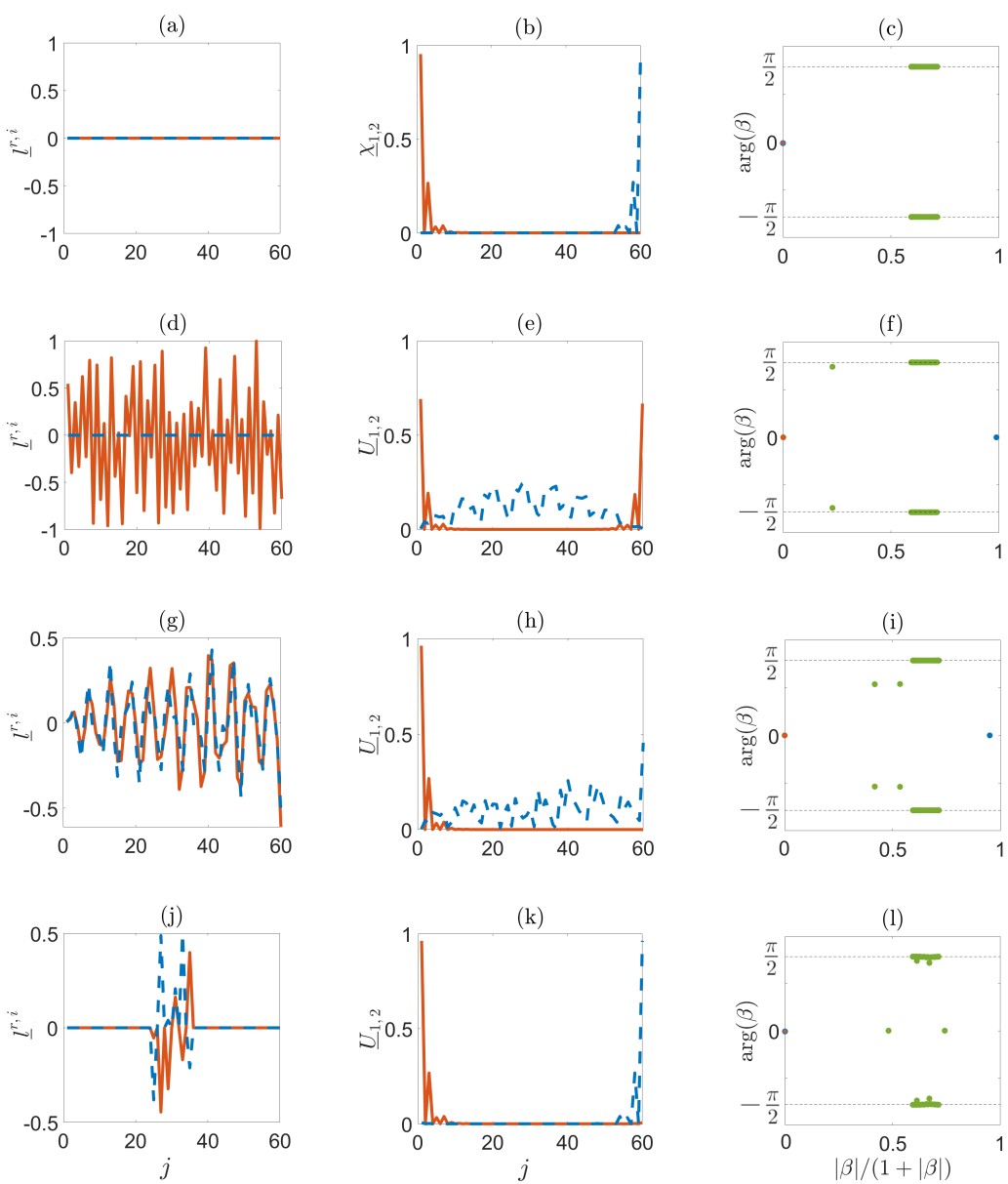

Figure 2: The effect of dissipative fields on the eigen functions and eigen values of $X$. Panels (a), (d), (g), (j): The spatial distribution of the dissipative fields $\underline{l}^r$ (red curve) and $\underline{l}^i$ (blue curve) along the chain. Panel (b): The spatial distribution of the MM wave function in the isolated system. Panel (c): The eigen values $\beta$ of the matrix $X$ for the isolated system. Panels (e), (h), (k): The right eigen vectors of the matrix $X$ corresponding to the eigen values with $\operatorname{Im}\beta = 0$ and minimal magnitudes $\operatorname{Re}\beta$. Panels (f), (i), (l): The eigen values $\beta$ of the matrix $X$. The first row corresponds to the isolated system while the next rows correspond to the dissipative fields chosen according to criteria (a), (b), and (c) in Sec. 4.4, respectively.

of dissipative fields in this case may be quite nontrivial, however, for the ZKM eigen vector is $\underline{U}_1 = \underline{\chi}_1$, since the dissipative field does not affect the MM with the wave function $\underline{\chi}_1$. An additional feature of the considered situation is the orthogonality of the dissipative fields to the subspace of the other gapped modes with a linear hull span$\{\underline{\chi}_7, \ldots, \underline{\chi}_{2N}\}$. Therefore, the eigen values of $X$ includes non-dissipating excitations with Re $\beta = 0$, which correspond to oscillating modes of the isolated system, cf. Fig. 2c and Fig. 2i. This situation corresponds to the fact that the Lindbladian spectrum contains a large number of zero modes, $\lambda_{\underline{\eta}} = 0$, see Eq. (28), and NESS is multiply degenerate.

The fourth row of Fig. 2 corresponds to the strategy of inducing ZKM, where the dissipative fields are chosen to be spatially structured and non-acting on the Majorana modes wave functions, see item (c) in Sec.4.4. In Fig. 2j, the dissipative fields were selected to affect the central part of the chain alone. Additionally, these dissipative fields were chosen so that they are nonzero at one Majorana sublattice, $l^{r,i}_{2j-1} \equiv 0$, and thus $\underline{l}^{r,i} \cdot \underline{\chi}_1 \equiv 0$, while the overlap with the wave function $\chi_2$ is exponentially suppressed, $\underline{l}^{r,i} \cdot \underline{\chi}_2 \sim e^{-L/\xi}$ (where $\xi \sim \Delta_1^{-1}$ - a superconducting coherence length). In this scenario, the spatial structure of stationary excitations is practically equivalent to the MM in the isolated system, $\underline{U}_1 = \underline{\chi}_1$ and $\underline{U}_2 \cong \underline{\chi}_2$ (cf. Fig. 2b and 2k). Thus, the open system exhibits a pair of kinetic modes with $\beta_1 = 0$ (ZKM) and $\beta_2 \cong 0$, corresponding to the red and blue dots near the origin in Fig. 2l. If the region of nonzero $\ell$ in Fig. 2j is shifted toward the localization area of $\chi_1$ (the left edge of the chain), the ZKM with $\beta_1 = 0$ and eigen vector $\underline{U}_1 = \underline{\chi}_1$ remains stable since $\ell \perp \underline{\chi}_1$. Conversely, if the center of mass of $\ell$ is shifted toward the localization of $\underline{\chi}_2$ (the right edge of the chain), the real part of $\beta_2$ begins to grow, while the eigen vector $\underline{U}_2$ starts to modify. With a sufficiently strong shift to the right, the spatial profiles and spectrum in Fig. 2k and 2l become analogous to those in Fig. 2h and 2i, respectively.

## 5.4 The spectrum dependence on dissipation strength: the case $N_M = 2$, $N_B = 1$:

Having discussed the mechanisms for inducing weak ZKMs, we now briefly examine the characteristic features of the system's spectral evolution as a function of dissipation strength. To this end, we study the spectrum of the matrix $X(\gamma) = A + \gamma M_{\boldsymbol{r}}$, which incorporates the dissipation intensity parameter $\gamma \in [0, 1]$. We consider the case with two pairs of MM in the isolated system ($N_M = 2$) coupled to a single bath ($N_B = 1$). In this case, there exist two robust ZKMs (see Eq. (55)) that remain invariant under dissipation: $\beta_{1,2}(\gamma) \equiv 0$. To visually distinguish these robust ZKMs from dissipation-sensitive modes in Fig. 3, we applied an infinitesimal energy shift $\beta_{1,2} \to \beta_{1,2} \pm i0^+$, represented by black points at $|\beta| = 0$ with phase angles arg $\beta = \pm\pi/2$.

Figure 3a displays the generic case where rk $\mathcal{B} = 2$ and $N_0 = 2$, in which the pair of modes with $\beta = 0$ is sensitive to dissipative perturbation. At small $\gamma$, these zero modes acquire positive real parts (indicating damping) while their degeneracy is lifted. As dissipation strength increases, an exceptional point is generated at a critical value $\gamma_c$, producing complex-conjugate eigenvalues that correspond to damped oscillatory modes (vertical spectral branches emerging after coalescence of red and blue dots in Fig. 3a). With further increase of $\gamma$, the dissipation significantly affects low-lying gapped modes, endowing them with substantial real components. These perturbed gapped modes exhibit two distinct behaviors: either maintaining complex-conjugate relationships or transitioning to real eigenvalues followed by splitting. The spectral reorganization occurs through the emergence of exceptional points.

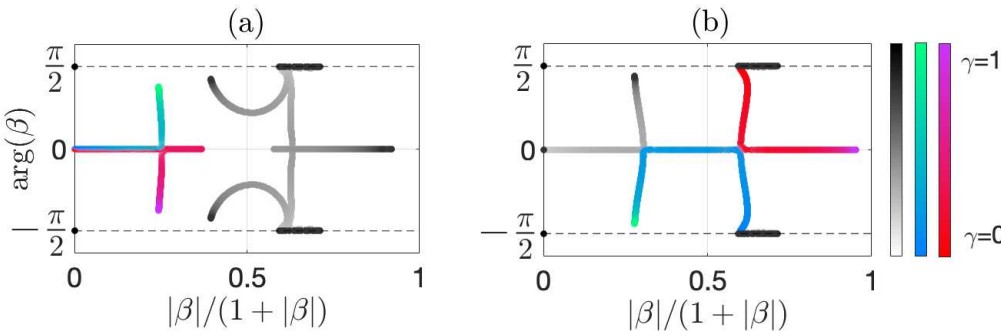

Figure 3: Spectral evolution as a function of the dissipation intensity $\gamma$, with color encoding $\gamma$-values according to the colorbar. (a) The case $\hat{L} \neq \hat{L}^+$, $\mathrm{rk}\,\mathcal{B} = 2$, $N_0 = 2$. (b) The case $\hat{L} = \hat{L}^+$, $\mathrm{rk}\,\mathcal{B} = 1$, $N_0 = 3$. The dissipation matrix $\gamma \underline{l}\,\underline{l}^\dagger$ governs the $\gamma$-dependence, while the other parameters match those in Fig. 2. Blue-green and red-purple colorbars correspond to spectral branches selected to demonstrate a role of exceptional points with a horizontal-to-vertical change of motion (and vice versa) under the increase of dissipation (the parameter $\gamma$).

If the dissipative bath has Hermitian jump operators ($L = L^+$, Fig. 3b), the system exhibits three stable zero modes (two robust ZKMs and one weak ZKM, as described in item (a) of Sec. 4.4). The dissipation shifts one mode along $\mathrm{Re}\,\beta$, while increasing $\gamma$ causes gapped modes to transition to real eigenvalues with $\partial\,\mathrm{Re}\,\beta/\partial\gamma$ of opposite signs (see evolution of the red and blue branches in Fig. 3b). This leads to the formation of an exceptional point through coalescence of the original zero mode and gapped mode (gray and blue branches behavior in Fig. 3b), resulting in complex eigenvalues and oscillatory behavior.

This behavior exemplifies non-Hermitian systems where perturbations create exceptional points and real-to-complex spectral transitions [114]. The parameter space near exceptional points is particularly noteworthy, though its analysis extends beyond this work.

## 5.5    Transfer of Majorana modes via bath hybridization

The effect of Majorana mode hybridization through a bath, described in Sec. 5.2 (see Eq. 76), enables control over ZKM by tuning dissipative fields. To illustrate such a control, let us consider a standard Kitaev chain with $N_r = 1$ in the topological phase, i.e. with $|\mu| < 2t$, hosting a single MM pair (the case $N_M = 1$). We assume that the system is coupled to a single dissipative field ($N_B = 1$), where the fermion pump and loss amplitudes satisfy $\underline{\mu} = e^{i\phi}\underline{\nu}$. Here, $\phi$ is a real parameter, while $\underline{\mu}, \underline{\nu} \in \mathbb{C}^N$, see Eq. (4).

Let us denote $\underline{\chi}_A$ and $\underline{\chi}_B \in \mathbb{R}^N$ the non-zero components of the Majorana wavefunctions $\underline{\chi}_1$ and $\underline{\chi}_2$, localized at odd and even sites of the chain, respectively: $\chi_{A,j} = \chi_{1,2j-1}$ and $\chi_{B,j} = \chi_{2,2j}$. Further, we define $\underline{\nu}^r = \mathrm{Re}\,\underline{\nu}$ and $\underline{\nu}^i = \mathrm{Im}\,\underline{\nu}$. Using Eq. (4), the hybridization matrix $\mathcal{B}$, see Eq. (53), becomes

$$\mathcal{B} = \frac{1}{2}\begin{pmatrix} (\underline{\chi}_A \cdot \underline{\nu}^r)\,(1 + \cos\phi) - (\underline{\chi}_A \cdot \underline{\nu}^i)\sin\phi & (\underline{\chi}_A \cdot \underline{\nu}^r)\sin\phi + (\underline{\chi}_A \cdot \underline{\nu}^i)\,(1 + \cos\phi) \\ -(\underline{\chi}_B \cdot \underline{\nu}^r)\sin\phi + (\underline{\chi}_B \cdot \underline{\nu}^i)\,(1 - \cos\phi) & -(\underline{\chi}_B \cdot \underline{\nu}^r)\,(1 - \cos\phi) - (\underline{\chi}_B \cdot \underline{\nu}^i)\sin\phi \end{pmatrix},$$
(78)

The determinant of the matrix $\mathcal{B}$ can be evaluated as

$$\det\mathcal{B} = -\frac{1}{2}\sin\phi \cdot \det\left[ \begin{pmatrix} \underline{\chi}_A \\ \underline{\chi}_B \end{pmatrix} \cdot \left(\underline{\nu}^r, \underline{\nu}^i\right) \right].$$
(79)

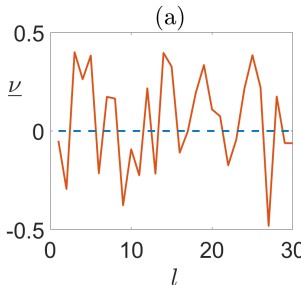 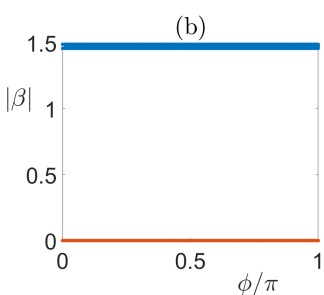 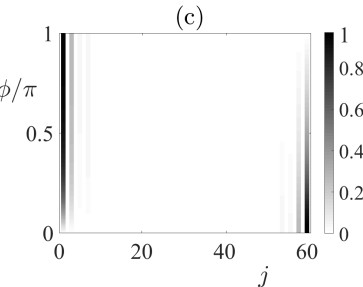

Figure 4: (a) Spatial profile of the gain field $\underline{\nu}$ related with the loss one $\underline{\mu}$ via $\underline{\mu} = e^{i\phi}\underline{\nu}$ (Eq. (4)). Solid and dashed lines show the real ($\underline{\nu}^r$) and imaginary ($\underline{\nu}^i$) components, respectively. (b) Phase dependence of kinetic mode energies $|\beta|$ on $\phi$. Red dots mark a weak zero kinetic mode present at all values of $\phi$. (c) Color map showing the $\phi$-dependent spatial distribution of the normalized ZKM weights across Majorana sublattices, see Eq. (80). Color scale indicates magnitude of $|U_j|(\phi)$. At $\phi = 0$ ($\phi = \pi$), the ZKM localizes near the right (left) chain edge. Intermediate $\phi$ values yield localization at the both edges due to bath-mediated Majorana hybridization. Other parameters match Fig. 2.

Therefore, the weak ZKM emerges if $\underline{\nu}^i = 0$ or $\sin\phi = 0$.

In the case $\underline{\nu}^i = 0$, the right eigen vector of the matrix $X$ corresponding to the weak ZKM is given as the linear combination of the MM wave functions:

$$\underline{U} \sim \underline{\chi}_1(\underline{\chi}_B \cdot \underline{\nu}^r)\sin\frac{\phi}{2} + \underline{\chi}_2(\underline{\chi}_A \cdot \underline{\nu}^r)\cos\frac{\phi}{2}. \tag{80}$$

The expression (80) demonstrates that as $\phi$ varies from 0 to $\pi$, the ZKM evolves continuously from $\underline{U}|_{\phi=0} \sim \underline{\chi}_2$ (the MM localized on right edge) to $\underline{U}|_{\phi=\pi} \sim \underline{\chi}_1$ (the MM localized on left edge). This evolution constitutes *the Majorana mode transfer* mediated by the hybridization with the bath. Importantly, for $\underline{\nu} \in \mathbb{R}^N$, such transfer from the right to the left chain occurs within a degenerate NESS ($\det\mathcal{B} \equiv 0$). The mechanism of the transfer appears rather straightforward. When $\underline{\mu} = \underline{\nu}$ ($\phi = 0$), the dissipative fields $\underline{l}^{r,i}$ act on the odd sites of the chain alone, maintaining $\overline{\ell} \perp \underline{\chi}_2$ and thus preserving the MM wave function $\underline{\chi}_2$. Conversely, when $\underline{\mu} = -\underline{\nu}$ ($\phi = \pi$), the dissipative fields selectively target the even sites of the chain, destroying the MM wave function $\underline{\chi}_2$ while leaving $\underline{\chi}_1$ to be unaffected.

We note that for $\underline{\nu} \in \mathbb{R}^N$ and $\phi \in [0, \pi]$, the dissipative field can be represented as:

$$\underline{l} = \underline{\tilde{l}}e^{i\phi/2}, \qquad \tilde{l}_{2j-1} = \nu_j^r\cos\frac{\phi}{2}, \qquad \tilde{l}_{2j} = -\nu_j^r\sin\frac{\phi}{2}. \tag{81}$$

In this case, the mechanism for inducing a weak ZKM naturally satisfies criterion (a) of Sec. 4.4. Furthermore, since $\underline{\tilde{l}} \in \mathbb{R}^{2N}$, the driving matrix $Y = \text{Im}\underline{l}\,\underline{l}^\dagger \equiv 0$. Consequently, the correlation matrix $F$ must satisfy the homogeneous Lyapunov equation $XF + FX^T = 0$. When regularization $X \to X_\epsilon = X + \epsilon UU^{-1}$ ($\epsilon \to 0_+$) is accounted for, we expect the NESS to exhibit $F = 0$, corresponding to a uncorrelated fermionic ensemble with $\langle c_l^+ c_m \rangle = \delta_{lm}/2$.

In the case $\underline{\nu} \in \mathbb{C}^N$, the weak ZKM exists only at $\phi = 0$ and $\phi = \pi$ while ZKM are absent at the intermediate magnitudes of $\phi$. Although, the initial and final stages of the protocol of changing $\phi$ from 0 to $\pi$ with $\underline{\nu} \in \mathbb{C}^N$ are the same as in the case $\underline{\nu} \in \mathbb{R}^N$, physically, it should be two very different situations.

In Figure 4 we illustrate the above results by means of numerical calculations. As shown in Fig. 4a for $\underline{\nu}^i = 0$, the system hosts a weak ZKM for arbitrary values of $\phi$ (Fig. 4b). The spatial distribution of the ZKM (80) evolves between $\underline{\chi}_1$ and $\underline{\chi}_2$ profiles as $\phi$ varies from 0 to $\pi$. At intermediate $\phi$ values, the $\underline{U}$ distribution becomes localized at both chain edges and across all Majorana sublattices.

# 6   Conclusions

To summarize, we considered the model of a topological superconductor affected by the dissipation not preserving the number of fermions. To study the system we apply the GKSL equation in which the unitary evolution is described by quadratic Hamiltonian, while jump operators are linear in fermionic creation and annihilation operators (linear dissipative fields). Analysing the dynamics within the GKSL equation, we addressed the following question: how the zero energy Majorana modes existing in the isolated system are affected by the dissipation transforming to the zero kinetic modes and under what conditions the number of ZKM can be maximized. We proved the exact relation for the number of the ZKM, see Eq. (57). We found that the eigenvectors of ZKM always span the same hull as the eigenvectors of MM, i.e. belonging to the kernel of the Hamiltonian, see Eq. (66). We proved that the existence of ZKM is related with the presence of the strong symmetry of the Lindbladian realized by the unitary operator, see Eq. (59). We estimated the number of distinct non-equilibrium steady states from below, see the end of Sec. 4.2. Based on the developed analytical results we proposed the practical recipes how to maximize the number of ZKM by controlling the dissipative fields, see Sec. 4.4.

We note that our results are applicable to the case when TS is coupled to reservoirs. We discussed under which conditions the coupling to reservoirs can be described in the framework of the dissipative fields.

We illustrated our results in the case of the Kitaev chain with long-range hoppings and superconducting pairings. We showed how one can use dissipation to manipulate the ZKM, e.g. transferring them from one edge of the chain to the other.

# Acknowledgements

We are grateful to A. Abanov, E. König, Yu. Makhlin, and A. Mel'nikov for useful discussions and comments. The authors acknowledge the hospitality during the international workshop "Landau Week 2025" where part of this work has been performed.

**Funding information**   The authors acknowledge personal support from the Foundation for the Advancement of Theoretical Physics and Mathematics "BASIS". The work of I.S.B. was partially supported by the Basic Research Program of HSE. The work was supported by the Ministry of Science and Higher Education of the Russian Federation (Project No. FFWR-2024-0017).

# 7 Appendix

In this Appendix we present details of derivation of the GKSL equation for the topological superconductor coupled to the reservoirs.

Following Refs. [115, 116], we derive the GKSL equation for the superconducting system tunnel coupled to metallic reservoirs. For a sake of simplicity we focus on the case of a single junction. Generalization to the case of several reservoirs is straightforward. The Hamiltonians of the superconducting system, $\hat{H}$, the reservoir, $\hat{H}_L$, and the tunneling junction, $\hat{V}$, read

$$\hat{H} = \sum_{\nu=1}^{N} \varepsilon_a \hat{\alpha}_a^+ \hat{\alpha}_a, \qquad \hat{H}_L = \sum_k \xi_k \hat{d}_k^+ \hat{d}_k$$

$$\hat{V} = \sum_{k,\,n} \left( t_{n,k} \hat{d}_k \hat{c}_n^+ + t_{n,k}^* \hat{c}_n \hat{d}_k^+ \right) = \sum_{n=1}^{N} \left( \hat{c}_n \hat{D}_n^+ + \hat{D}_n \hat{c}_n^+ \right). \tag{82}$$

Here we introduce $\hat{D}_n = \sum_k t_{n,k} \hat{d}_k$. The density matrix of the total system evolves in accordance with the following equation (here and further $\tilde{x}$ denotes operator $\hat{x}$ in the interaction picture):

$$\dot{\tilde{\rho}}_T(t) = -i \left[ \tilde{V}(t), \, \tilde{\rho}_T(t) \right], \qquad \tilde{\rho}_T(t) = \tilde{\rho}_T(-\infty) - i \int_{-\infty}^{t} ds \left[ \tilde{V}(s), \, \tilde{\rho}_T(s) \right]. \tag{83}$$

In the interaction representation the operators become time-dependent. In particular, we find

$$\tilde{c}_n(t) = \sum_{\nu=1}^{N} \left( u_{\nu n}^* e^{-i\varepsilon_\nu t} \hat{\alpha}_\nu + v_{\nu n} e^{i\varepsilon_\nu t} \hat{\alpha}_\nu^+ \right), \quad \tilde{D}_n(t) = \sum_k t_{n,k} \hat{d}_k e^{-i\xi_k t}. \tag{84}$$

As usual, we assume that the tunneling Hamiltonian switches adiabatically at $t = -\infty$ such that $\tilde{V}(-\infty) = 0$. Tracing Eq. (83) over the reservoir's degrees of freedom, we obtain the evolution equation for the reduced density matrix $\tilde{\rho}(t)$ of the superconducting system as follows

$$\dot{\tilde{\rho}}(t) = -i \operatorname{Tr}_E \left( \left[ \tilde{V}(t), \, \hat{\rho}(-\infty) \otimes \hat{\rho}_E(-\infty) + \hat{\rho}_{corr}(-\infty) \right] \right) -$$

$$- \int_{-\infty}^{t} ds \operatorname{Tr}_E \left( \left[ \tilde{V}(t), \left[ \tilde{V}(s), \, \tilde{\rho}(t) \otimes \tilde{\rho}_E(t) + \tilde{\rho}_{corr}(t) \right] \right] \right) + \mathcal{O}(\hat{V}^3). \tag{85}$$

Here we introduced the density matrix $\tilde{\rho}_{corr}(t)$ describing correlations of the superconductor and the reservoir in addition to the factorized density matrix $\tilde{\rho}(t) \otimes \tilde{\rho}_E(t)$.

Further analysis of Eq. (85) is performed within the following set of the assumptions [6]:

(i) Assumption of *absence of correlation between the system and the reservoir* in the infinite past: $\hat{\rho}_{corr}(-\infty) = 0$. Additionally, we assume that the reservoir is in the thermal equilibrium with some temperature $T$: $\hat{\rho}_E(-\infty) = \hat{\rho}_E^{(eq)}$. As a corollary, the first term in the right hand side of Eq. (85) vanishes.

---

[6]Note that we do not employ the rotating wave approximation which is a standard choice, see e.g. Ref. [115, 116]. However, in our opinion, the rotating wave approximation is too restrictive and even may be not applicable in the case of zero energy MM.

(ii) Assumption of *weak tunnel coupling*, $\hat{V} \ll 1$. It allows us to limit consideration by the second order perturbation theory in $V$ (essentially, to work on the level of the Fermi's Golden rule). In addition, the smallness of $\hat{V}$ justifies the slow dynamics of the reduced density matrix.

(iii) Assumption of *separation of time scales*: the characteristic time scale of the evolution of the superconductor $\tau$ is much longer than the correlation time, $\tau_{corr}$, characterized correlations between the system and reservoir as well as the relaxation time, $\tau_E$, in the reservoir, i.e. $\tau \sim 1/V^2 \gg \tau_E, \tau_{corr}$. As a corollary, the reduced density matrix becomes $\tilde{\rho}(t) = \tilde{\rho}(t) \otimes \tilde{\rho}_E(t) + \tilde{\rho}_{corr}(t) \cong \tilde{\rho}(t) \otimes \hat{\rho}_E^{(eq)} \Big|_{t \sim \tau}$.

Using the assumptions (i)-(iii), we obtain (we use substitution $s \to t - s$):

$$
\dot{\tilde{\rho}}(t) = \int_0^\infty ds\, \mathrm{Tr}_E \left( \left[ \tilde{V}(t), \left[ \tilde{V}(t-s),\ \tilde{\rho}(t) \otimes \hat{\rho}_E^{(eq)} \right] \right] \right) = \tag{86}
$$

$$
= \int_0^\infty ds \sum_{n,m=1}^N \left( \left[ \tilde{c}_n(t),\, \tilde{c}_m^+(t-s)\, \tilde{\rho}(t) \right] \cdot \mathrm{Tr}_E \left( \tilde{D}_n^+(t)\, \tilde{D}_m(t-s)\, \hat{\rho}_E^{(eq)} \right) + \right.
$$

$$
\left. + \left[ \tilde{c}_n^+(t),\, \tilde{c}_m(t-s)\, \tilde{\rho}(t) \right] \cdot \mathrm{Tr}_E \left( \tilde{D}_n(t)\, \tilde{D}_m^+(t-s)\, \hat{\rho}_E^{(eq)} \right) + h.c. \right).
$$

The assumption (iii) implies that the correlation function of the reservoir involved in Eq. (86) depend on time $s$ only:

$$
\mathrm{Tr}_E \left( \tilde{D}_n^+(t)\, \tilde{D}_m(t-s)\, \hat{\rho}_E^{(eq)} \right) = \mathrm{Tr}_E \left( \tilde{D}_n^+(s)\, \hat{D}_m\, \hat{\rho}_E^{(eq)} \right) = \sum_k t_{n,k}^*\, t_{m,k}\, e^{i\xi_k s} f(\xi_k), \tag{87}
$$

$$
\mathrm{Tr}_E \left( \tilde{D}_n(t)\, \tilde{D}_m^+(t-s)\, \hat{\rho}_E^{(eq)} \right) = \mathrm{Tr}_E \left( \tilde{D}_n(s)\, \hat{D}_m^+\, \hat{\rho}_E^{(eq)} \right) = \sum_k t_{n,k}\, t_{m,k}^*\, e^{-i\xi_k s} f(-\xi_k). \tag{88}
$$

Making inverse transformation to the Schrödinger picture, $\tilde{\rho}(t) \to \hat{\rho}(t) = e^{-iHt}\, \tilde{\rho}\, e^{iHt}$, we find the following equation for the reduced density matrix

$$
\dot{\hat{\rho}}(t) = -i \left[ \hat{H},\ \hat{\rho}(t) \right] - \mathcal{D}[\hat{\rho}], \tag{89}
$$

$$
\mathcal{D}[\hat{\rho}] = \int_0^\infty ds \left( \left[ \tilde{c}_m^+(-s)\, \hat{\rho}(t),\, \hat{c}_n \right] \langle \tilde{D}_n^+(s)\, \hat{D}_m \rangle + \left[ \tilde{c}_m(-s)\, \hat{\rho}(t),\, \hat{c}_n^+ \right] \langle \tilde{D}_n(s)\, \hat{D}_m^+ \rangle + h.c. \right).
$$

Integrating over time $s$, we obtain

$$
\mathcal{D}[\hat{\rho}] = \sum_{n,m,l=1}^N \sum_{a=1}^N \left( \overbrace{D_{nm}^{(+)}(\varepsilon_a)\, A_{ml;a}}^{\Gamma_{nl}^{(ee)}(\varepsilon_a)} \cdot \left[ \hat{c}_l^+\, \hat{\rho},\, \hat{c}_n \right] + \overbrace{D_{nm}^{(-)}(\varepsilon_a)\, A_{ml;a}^*}^{\Gamma_{nl}^{(hh)}(\varepsilon_a)} \cdot \left[ \hat{c}_l\, \hat{\rho},\, \hat{c}_n^+ \right] + \right.
$$

$$
\left. + \underbrace{D_{nm}^{(-)}(\varepsilon_a)\, B_{ml;a}^*}_{\Gamma_{nl}^{(eh)}(\varepsilon_a)} \cdot \left[ \hat{c}_l^+\, \hat{\rho},\, \hat{c}_n^+ \right] + \underbrace{D_{nm}^{(+)}(\varepsilon_a)\, B_{ml;a}}_{\Gamma_{nl}^{(he)}(\varepsilon_a)} \cdot \left[ \hat{c}_l\, \hat{\rho},\, \hat{c}_n \right] + h.c. \right). \tag{90}
$$

Here we introduced the matrices

$$
A_{ml;a} = u_{am} u_{al}^* + v_{am}^* v_{al}, \quad B_{ml;a} = u_{am} v_{al}^* + v_{am}^* u_{al}, \tag{91}
$$

as well as the matrices encoding the coupling between the superconductor and the reservoir:

$$D_{nm}^{(\pm)}(\varepsilon_a) = \sum_k \left( t_{n,k}^* \, t_{m,k} \right)^{\zeta_\pm} f(\pm \xi_k) \int_0^\infty ds \left( e^{\pm i(\xi_k + \varepsilon_a)s} + e^{\pm i(\xi_k - \varepsilon_a)s} \right) =$$

$$= \sum_k \left( t_{n,k}^* \, t_{m,k} \right)^{\zeta_\pm} f(\pm \xi_k) \left[ \pi \left( \delta(\xi_k + \varepsilon_a) + \delta(\xi_k - \varepsilon_a) \right) \pm i \, \text{p.v.} \left( \frac{1}{\xi_k + \varepsilon_a} + \frac{1}{\xi_k - \varepsilon_a} \right) \right], \quad (92)$$

where $\zeta_+ = 1$, $\zeta_- = *$ (complex conjugation operation). Next, we introduce Nambu representation:

$$\hat{\Psi} = \left( \begin{array}{c} \underline{\hat{c}} \\ \underline{\hat{c}}^+ \end{array} \right), \ \Gamma = \sum_{a=1}^N \left( \begin{array}{cc} \Gamma^{(ee)}(\varepsilon_a) & \Gamma^{(eh)}(\varepsilon_a) \\ \Gamma^{(he)}(\varepsilon_a) & \Gamma^{(hh)}(\varepsilon_a) \end{array} \right), \ \gamma = \frac{1}{2} \left( \Gamma + \Gamma^+ \right), \ \kappa = \frac{1}{2i} \left( \Gamma - \Gamma^+ \right), (93)$$

where $\gamma = \gamma^+ \in \mathbb{C}^{2N \times 2N}$ and $\kappa = \kappa^+ \in \mathbb{C}^{2N \times 2N}$. Then the reduced density matrix is evolved in accordance with the following equation:

$$\dot{\hat{\rho}}(t) = -i \left[ \hat{H} + \hat{H}_{LS}, \ \hat{\rho}(t) \right] + \mathcal{D}[\hat{\rho}], \quad (94)$$

$$\hat{H}_{LS} = \sum_{i,j=1}^{2N} \kappa_{ij} \, \hat{\Psi}_i \, \hat{\Psi}_j^+, \quad \mathcal{D}[\hat{\rho}] = \sum_{i,j=1}^{2N} \gamma_{ij} \left( 2 \, \hat{\Psi}_i^+ \hat{\rho} \, \hat{\Psi}_j - \{ \hat{\rho}, \, \hat{\Psi}_i \, \hat{\Psi}_j^+ \} \right). \quad (95)$$

The Hamiltonian $\hat{H}_{LS}$, the so-called Lamb shift, is the correction to the unitary dynamics due to virtual transitions of fermions from the superconductor to the reservoir and back. The term $\mathcal{D}[\hat{\rho}]$ describes the dissipative effects due to coupling to the reservoir. In order to transform it to the standard GKSL form, (1), we benefited from the eigenbasis of the matrix $\gamma$:

$$\mathcal{D}[\hat{\rho}] = \sum_{v=1}^{\text{rk}\,\gamma} \left( 2\hat{L}_v \hat{\rho} \hat{L}_v^+ - \hat{L}_v^+ \hat{L}_v \hat{\rho} - \hat{\rho} \hat{L}_v^+ \hat{L}_v \right), \qquad \hat{L}_v = \sqrt{\lambda_v} \cdot \sum_{l=1}^N \left( \Omega_{v,N+l}^* \, c_l + \Omega_{v,l}^* \, c_l^+ \right),$$

$$\gamma = \Omega^+ \bar{\gamma} \, \Omega, \qquad \bar{\gamma} = \text{diag} \left( \lambda_1, \, \ldots, \, \lambda_{2N} \right). \quad (96)$$

Here we assume that $\sqrt{\lambda_v} \in \mathbb{R}_+$ or $i\,\mathbb{R}_+$. Therefore, tunnel junction between the superconductor and the reservoir results in the GKSL equation with jump operators $\hat{L}_v$ which are linear in the fermionic operators. We note that the reservoir is equivalent to $\text{rk}\,\gamma$ dissipative baths.

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
