# Peer review of "Dissipation-Induced Steady States in Topological Superconductors: Mechanisms and Design Principles"

_SciPost Physics_

## Round 1 · Referee Report · Anonymous (Referee 1) · 2025-9-30

Strengths

1) Establishes a novel algebraic relation between equilibrium Majorana zero modes and dissipative kinetic zero modes. 2) Provides a transparent symmetry-based framework via GKSL + third quantization. 3) Demonstrates concepts concretely on a generalized BDI-class Kitaev chain with numerics. 4) Addresses a timely question of dissipative engineering of topological steady states.

Weaknesses

1) Algebraic relation not stated with full rigor; assumptions remain not fully clear. 2) Robustness to realistic perturbations, i.e., interactions, disorder, non-Markovianity insufficiently addressed. 3) Limited comparison to prior no-go theorems and related results.

Report

The manuscript investigates how dissipative dynamics can generate degenerate nonequilibrium steady states in topological superconductors. Using the Lindblad (GKSL) framework and third-quantization techniques (vectorisation of the density matrix), the authors identify a correspondence between equilibrium Majorana zero modes and dissipative “kinetic zero modes.” They derive a compact algebraic relation connecting the number of these modes to hybridization matrices describing the overlap of single-particle states with dissipative fields.

This is applied to a generalized Kitaev chain (class BDI with long-range couplings), and the authors illustrate how appropriate Lindblad operators can stabilize steady-state degeneracies.
The work is timely and relevant, and the analytic–numerical combination is a strength. The algebraic relation is useful to both theorists and experimentalists interested in dissipative state engineering. However, several aspects require clarification and strengthening before the results can be fully appreciated and trusted.

First, the algebraic counting relation could be stated with greater rigor: its assumptions, scope, and precise conditions of validity should be formulated explicitly (ideally as a theorem with proof in an appendix).

Second, robustness issues are only lightly touched upon. Since realistic systems inevitably involve interactions, disorder, and non-Markovian baths, the authors should analyze or at least discuss stability under such perturbations.

Third, the connection to prior literature, particularly “no-go” results on dissipative topology, should be sharpened: the manuscript must explicitly delineate how the present approach circumvents or complements those limitations.

In summary, this is a promising and novel contribution which could merit publication in SciPost Physics after minor revision. With clearer formulation of the central relation and an expanded discussion of robustness, prior work, and experimental feasibility, the manuscript would reach a greater level of clarity and reliability, which would support publication in SciPost Physics.

Requested changes

1) Formulate the counting relation precisely (theorem-style, with explicit assumptions). 2) Discuss robustness to perturbations (disorder, interactions, non-Markovian effects). 3) Sharpen the discussion of prior no-go results and clearly position the present work.

Recommendation

Ask for minor revision

---

## Editorial Decision

in_refereeing